# Post-phagocytosis activation of NLRP3 inflammasome by two novel T6SS effectors

**Hadar Cohen, Noam Baram, Chaya Mushka Fridman, Liat Edry-Botzer, Dor Salomon\*, Motti Gerlic\***

Department of Clinical Microbiology and Immunology, Sackler Faculty of Medicine, Tel Aviv University, Tel Aviv, Israel

**Abstract** The type VI secretion system (T6SS) is used by bacteria to deliver toxic effectors directly into target cells. Most T6SSs mediate antibacterial activities, whereas the potential anti-eukaryotic role of T6SS remains understudied. Here, we found a *Vibrio* T6SS that delivers two novel effectors into mammalian host immune cells. We showed that these effectors induce a pyroptotic cell death in a phagocytosis-dependent manner; we identified the NLRP3 inflammasome as being the underlying mechanism leading to the T6SS-induced pyroptosis. Moreover, we identified a compensatory T6SS-induced pathway that is activated upon inhibition of the canonical pyroptosis pathway. Genetic analyses revealed possible horizontal spread of this T6SS and its anti-eukaryotic effectors into emerging pathogens in the marine environment. Our findings reveal novel T6SS effectors that activate the host inflammasome and possibly contribute to virulence and to the emergence of bacterial pathogens.

## Editor's evaluation

This study reports the function of novel effectors of the type VI secretion system (T6SS) of *Vibrio proteolyticus*, a *Vibrio* isolated from corals. The significance of the findings also extends to the identification of the mechanism by which these effectors induce pyroptotic cell death in mammalian host immune cells. Overall, the reported findings further contribute to the understanding of virulence mechanisms of bacterial pathogens.

**\*For correspondence:**
dorsalomon@mail.tau.ac.il (DS);
mgerlic@tauex.tau.ac.il (MG)

**Competing interest:** The authors declare that no competing interests exist.

## Introduction

Innate immune responses combat infections but may also drive pathological inflammation. The innate immune system engages an array of pattern-recognition receptors (PRR) that are expressed by cells found at the defensive front line against infections including macrophages, dendritic cells, neutrophils, and others. PRRs can detect a variety of microbial determinants termed pathogen-associated molecular patterns (PAMPs). Moreover, damaged host cells can also trigger PRRs by releasing danger-associated molecular patterns (DAMPs) (*Wallach et al., 2014*; *Wen et al., 2013*).

Some PRRs of the innate immune system are assembled into a high-molecular weight complex called 'the inflammasome' after sensing DAMPs or PAMPs. Inflammasome complexes are considered central components of innate immunity because of their ability to kill an infected cell (a cell death process that is termed pyroptosis), or by their activation and the subsequent secretion of mature proinflammatory cytokines interleukin 1β (IL-1β) and IL-18 (*Lamkanfi and Dixit, 2014*; *Schroder and Tschopp, 2010*). Inflammasomes can be divided into: (1) a canonical pathway in which Caspase-1 is cleaved into a catalytically active enzyme, and (2) non-canonical pathways in which Caspase-11 is activated (*Lamkanfi and Dixit, 2014*). The canonical inflammasome comprises one of the nucleotide-binding

oligomerization domain (NOD), leucine-rich repeat (LRR)-containing protein (NLR) family members, including NLRP1, NLRP3, NLRC4, or the DNA sensor absent in melanoma 2 (AIM2), all of which contain either a pyrin domain (PYD) or a caspase recruitment domain (CARD) (*Schroder and Tschopp, 2010*). These domains interact with apoptosis-associated speck-like protein (ASC), leading to its polymerization into large helical filaments known as ASC specks by facilitating self-interactions of the PYD of ASC (*Franklin et al., 2014*; *Proell et al., 2013*). The CARD domain of the ASC specks recruits Caspase-1 via a CARD-CARD interaction, leading to Caspase-1 clustering that permits auto-cleavage and formation of the active Caspase-1 p10/p20 tetramer (*Lamkanfi and Dixit, 2014*). Activated Caspase-1 processes pro-IL-1β and pro-IL-18 into their mature forms. Simultaneously, activated Caspase-1 cleaves gasdermin D (GSDMD) into a C-terminal fragment (GSDMD-CT) and N-terminal fragment (GSDMD-NT) (*Liu et al., 2016*; *Shi et al., 2015*). GSDMD-NT oligomerizes in membranes to form pores which promote IL-1β release and pyroptotic cell death (*Lieberman et al., 2019*; *Liu et al., 2016*).

The type VI secretion system (T6SS) is a widespread macromolecular protein secretion apparatus that translocates proteins into neighboring cells in a contact-dependent manner (*Bingle et al., 2008*; *Boyer et al., 2009*; *Mougous et al., 2006*; *Pukatzki et al., 2006*). Most of the T6SS-secreted proteins are toxic effectors capable of manipulating target cells. Although many T6SS effectors are antibacterial toxins, some T6SSs deliver effectors with anti-eukaryotic activities into host cells, thus promoting virulence or defense against predation (*Monjarás Feria and Valvano, 2020*). Nevertheless, the anti-eukaryotic potential of T6SSs remains understudied and possibly underappreciated (*Dar et al., 2018*; *Hachani et al., 2016*).

T6SS is common in vibrios (*Dar et al., 2018*), a genus of Gram-negative marine bacteria that includes many established and emerging human pathogens (e.g. *Vibrio cholerae*, *Vibrio parahaemolyticus,* and *Vibrio vulnificus*), which are a leading cause of gastroenteritis and wound infections (*Baker-Austin et al., 2018*). Vibrios preferentially grow in warm (>15°C) marine environments; remarkably, the rise in sea surface temperatures correlates with the spread of vibrios and with *Vibrio*-associated human illness (*Le Roux et al., 2015*; *Vezzulli et al., 2016*).

*Vibrio proteolyticus* (*V. proteolyticus*) is a Gram-negative marine bacterium that was originally isolated from the intestine of the wood borer *Limnoria tripunctata* (*Merkel et al., 1964*); it was also isolated from diseased corals (*Cervino et al., 2008*) and was shown to cause mortality in fish (*Bowden et al., 2018*) and in the crustacean model organism, *Artemia* (*Verschuere et al., 2000*). *V. proteolyticus* does not possess a type III secretion system (T3SS) that can mediate delivery of effector proteins directly into host cells (*Miller et al., 2019*; *Ray et al., 2016*). Nevertheless, it encodes a secreted pore-forming hemolysin (*Cohen et al., 2020*; *Ray et al., 2016*) and three different T6SSs (T6SS1-3) (*Ray et al., 2016*; *Ray et al., 2017*). T6SS1 mediates both antibacterial and anti-eukaryotic activities, whereas T6SS3 does not play any role in bacterial competition and its hypothesized anti-eukaryotic potential remains to be investigated (*Ray et al., 2017*).

In a recent study in which we investigated the secreted *V. proteolyticus* cytotoxic hemolysin, VPRH, we observed the existence of another, VPRH-independent cell death that occurred only during infections of primary macrophages, but not in the cultured cell lines HeLa or RAW264.7 (*Cohen et al., 2020*). Thus, in this study, we set out to determine whether *V. proteolyticus* T6SSs can lead to cell death in primary macrophages, and if so, to decipher the underlying mechanism. We found that *V. proteolyticus* T6SS3, but not T6SS1, induced a phagocytosis-dependent cell death in bone marrow-derived macrophages (BMDMs). Using chemical and genetics approaches, we identified pyroptosis as the underlying mechanism of T6SS3-induced cell death, and specifically the NLRP3 inflammasome. Moreover, we found that a pathway comprising gasdermin E (GSDME) and caspase-3 compensates for GSDMD absence in response to T6SS3 activity, in a NLRP3-dependent manner. Last, we identified a new T6SS3 activator and two novel T6SS3 anti-eukaryotic effectors that are needed to induce inflammasome-mediated cell death in BMDMs. Overall, our findings shed new light on the interplay between T6SS, phagocytosis and the NLRP3 inflammasome, and underscore the virulence potential of T6SSs.

## Results

### T6SS3 induces cell death in primary macrophages

We previously showed that *V. proteolyticus* induces two types of cell death in BMDMs: (1) a rapid cell death that is mediated by the pore-forming hemolysin, VPRH, and (2) a slower cell death that is VPRH-independent (*Cohen et al., 2020*). The underlying mechanism driving the latter remains unknown. *V. proteolyticus* harbors three T6SSs (T6SS1-3). We hypothesized that one or more of these T6SSs are responsible for the observed VPRH-independent cell death. Previously, we showed that T6SS1 mediates interbacterial competition and manipulates the actin cytoskeleton in macrophages (*Ray et al., 2017*). We also proposed that T6SS3 targets eukaryotic cells (*Ray et al., 2017*). Furthermore, we reported that T6SS1 and T6SS3 are functional under laboratory conditions; however, we did not detect any T6SS2 activating conditions (*Ray et al., 2017*). Therefore, we excluded T6SS2 from subsequent analyses, and we focused on a possible contribution of T6SS1 and T6SS3 to cell death in BMDMs.

To investigate the role of T6SS1 and T6SS3 in cell death, we first sought to further induce T6SS activity. We reasoned that if a T6SS is responsible for inducing cell death, then hyper-activating it would improve our ability to monitor the cell death phenotype by (1) maximizing its amplitude, (2) shortening the time to onset, and (3) allowing us to lower the multiplicity of infection (MOI). We previously reported that H-NS, the histone-like nucleoid-structuring protein, is a negative regulator of T6SSs in vibrios and that its deletion results in hyper-activation of T6SS (*Salomon et al., 2014*; *Salomon et al., 2015*). Thus, we generated a *V. proteolyticus* strain with a deletion in a gene we named *hns1* (*VPR01S_RS04360*); it encodes an H-NS that is very similar to VP1133, the previously studied H-NS protein from *V. parahaemolyticus* (*Salomon et al., 2014*). Importantly, the deletion was performed in a Δ*vprh* background to allow the subsequent monitoring of VPRH-independent cell death. We then determined the effect of *hns1* deletion on T6SS1 and T6SS3 activity. To this end, we monitored the expression and secretion of VgrG1 and Hcp3, hallmark secreted tail tube components of T6SS1 and T6SS3, respectively. We found that *hns1* deletion resulted in elevated secretion of both VgrG1 and Hcp3, compared to their secretion from the parental Δ*vprh* strain (*Figure 1a*). Notably, the secretion of VgrG1 and Hcp3 was T6SS1- and T6SS3-dependent, respectively, as evident from their absence in the supernatant fractions of strains deleted for *tssG1* (T6SS1⁻) and *tssL3* (T6SS3⁻), which are core structural genes required for the activity of T6SS1 and T6SS3, respectively. These results indicate that *hns1* deletion derepresses T6SS1 and T6SS3, leading to their constitutive activation and thus elevated secretion.

Next, we used real-time microscopy (IncucyteZOOM) to monitor VPRH-independent cell death kinetics in BMDMs. Whereas, infection with the Δ*vprh* strain resulted in negligible cell death 3 hr post infection, the Δ*vprh*/Δ*hns1* mutant induced a rapid cell death in which ~80% of the cells in the culture were dead within this time frame (set as 100% Normalized Area Under the Curve [AUC]; *Figure 1b and c*). Remarkably, the rapid cell death evident in the Δ*vprh*/Δ*hns1* mutant was completely abrogated upon genetic inactivation of T6SS3 (Δ*vprh*/Δ*hns1*/T6SS3⁻), but not upon genetic inactivation of T6SS1 (Δ*vprh*/Δ*hns1*/T6SS1⁻) (*Figure 1b and c*). Notably, genetic inactivation of T6SS1 or T6SS3 had no significant effect on growth during the relevant time frame compared to the parental Δ*vprh*/Δ*hns1* strain in either marine lysogeny broth (MLB) or Dulbecco's Modified Eagle Medium (DMEM) media (*Figure 1d*), nor did it affect bacterial swimming motility (*Figure 1e*). Taken together, these results suggest that T6SS3, but not T6SS1, mediates a VPRH-independent cell death in BMDMs.

### T6SS3 induces pyroptotic cell death in BMDMs

We previously demonstrated that a VPRH-independent cell death is observed upon infection of BMDMs, but not in HeLa cells or RAW 264.7 macrophages (*Cohen et al., 2020*). Unlike the two cell lines, BMDMs contain a functional cell death mechanism known as pyroptosis (*Cohen et al., 2020*; *Proell et al., 2013*). Pyroptosis is a form of necrotic and inflammatory regulated cell death induced by the inflammasome complex. Assembly of the inflammasome leads the maturation and secretion of pro-inflammatory cytokines, such as IL-1β (*Man et al., 2017*). Therefore, we hypothesized that inflammasome activation and pyroptosis are the mechanisms underlying the VPRH-independent, T6SS3-dependent cell death in BMDMs. To test this hypothesis, we monitored the effect of specific inflammasome inhibitors on T6SS3-mediated cell death. Using real-time microscopy, we observed that addition of either inhibitors, MCC950 (blocks ASC oligomerization by inhibiting the canonical

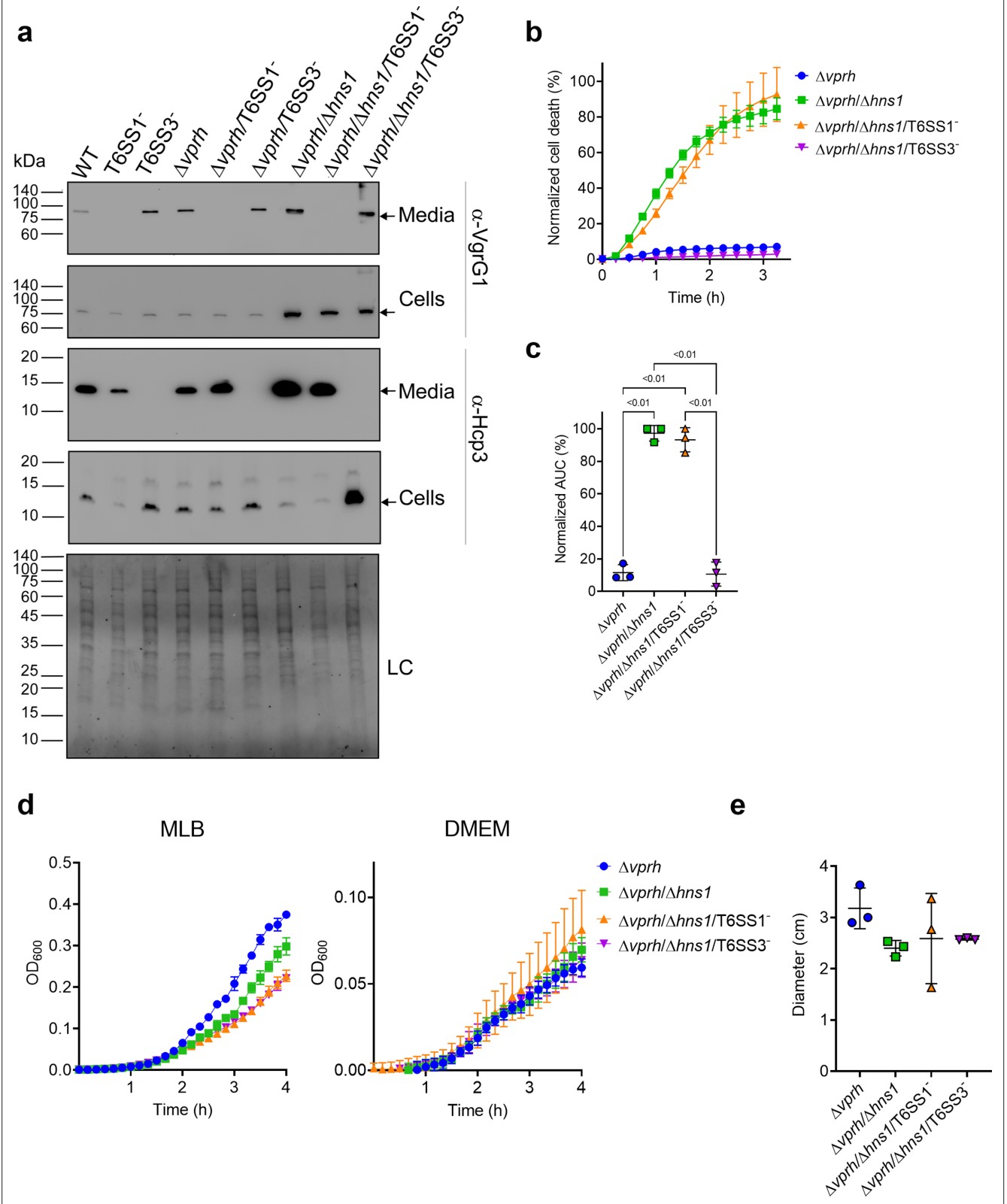

**Figure 1.** T6SS3 induces cell death in primary macrophages. (**a**) Expression (cells) and secretion (media) of Hcp3 and VgrG1 from the indicated *V. proteolyticus* strains were detected by immunoblotting using specific antibodies against Hcp3 and VgrG1, respectively. Loading control (LC) is shown for total protein lysate. Arrows denote the expected band size. (**b–c**) Assessment of cell death upon infection of bone marrow-derived macrophages (BMDMs) *with V. proteolyticus* strains. Approximately 3.5×10⁴ BMDMs were seeded into 96-well plates in triplicate and were primed with

*Figure 1 continued on next page*

Figure 1 continued

lipopolysaccharides (LPS) (100 ng/mL) 3 hr prior to infection with *V. proteolyticus* strains at multiplicity of infection (MOI) 5. Propidium iodide (PI) was added to the medium prior to infection, and its kinetic uptake was assessed using real-time microscopy (IncucyteZOOM) (**b**) and then analyzed as the area under the curve (AUC) of the percentage of PI-positive cells normalized to the number of cells in the wells (**c**). (**d**) Growth of *V. proteolyticus* strains, used in (**b**), in marine lysogeny broth (MLB) or Dulbecco's Modified Eagle Medium (DMEM) media at 30 or 37°C, respectively, measured as absorbance at 600 nm ($OD_{600}$). (**e**) Swimming motility of *V. proteolyticus* strains used in (**b**), measured as migration of a soft-agar plate after overnight incubation at 30°C. The data in (**a, b, d**) are a representative experiment out of three independent experiments. The data in (**c**) and (**e**) are the combined results of three independent experiments presented as the mean ± SD. In (**c**) and (**e**), statistical comparisons between the different *V. proteolyticus* strains were performed using a one-way ANOVA, followed by Tukey's multiple comparison test. Significant p-values (<0.05) are denoted above.

The online version of this article includes the following source data for figure 1:

**Source data 1.** Immunoblots of V. proteolyticus Hcp3 and VgrG1 expression and secretion.

and non-canonical NLRP3 inflammasome *Shao et al., 2015*) or Vx765 (a potent and selective competitive inhibitor of Caspase-1 *Church et al., 2008*) resulted in delayed and reduced cell death during infection with Δ*vprh*/Δ*hns1* bacteria (~40% reduction as measured by calculating the AUC), compared with a dimethyl sulfoxide (DMSO)-treated control (*Figure 2a and b*). As expected, inactivation of T6SS1 had no effect on Δ*vprh*/Δ*hns1*-induced cell death nor on the protective effect of the inhibitors, whereas inactivation of T6SS3 abrogated the cell death phenotype. These results support our hypothesis that T6SS3 induces cell death in BMDMs via an inflammasome-dependent pyroptotic pathway.

Furthermore, support for this hypothesis was obtained by monitoring the cleavage and secretion of the inflammasome-dependent cytokine, IL-1β, upon infection with *V. proteolyticus* strains. As shown in *Figure 2c–e*, IL-1β cleavage and secretion were dependent on the presence of a functional T6SS3 in Δ*vprh*/Δ*hns1* strains, and they were inhibited (*Figure 2c*) or undetected (*Figure 2d and e*) when inflammasome inhibitors were added. However, secretion of TNFα (an NF-κB-dependent, inflammasome-independent cytokine) was independent of T6SS3 activity and was unaffected by addition of inflammasome inhibitors (*Figure 2c*). Additional hallmark pyroptotic processes, such as the cleavage and release of Caspase-1 and GSDMD, were apparent upon infection of BMDMs with the Δ*vprh*/Δ*hns1* strain; however, they were undetected upon inactivation of T6SS3 or upon the addition of inflammasome inhibitors (*Figure 2d and e*). Taken together, these results suggest that T6SS3 induces an inflammasome-dependent, pyroptotic cell death in BMDMs.

## T6SS3 activates the NLRP3 inflammasome in BMDMs

Different members of the NLR family, including NLRP1 and NLRP3, can induce inflammasome assembly and activation, leading to pyroptotic cell death (*Bauernfried et al., 2021*; *Lamkanfi and Dixit, 2014*). NLRP3 is the most studied inflammasome-activating NLR; it is activated during bacterial, viral, and fungal infections, as well as in sterile inflammation mediated by endogenous DAMPs (*Swanson et al., 2019*). To identify the specific inflammasome pathway that is activated by T6SS3 in BMDMs, we generated BMDMs from knock-out (KO) mice in different inflammasome pathways (i.e., *Nlrp1*[-/-] and *Nlrp3*[-/-]) and infected them with *V. proteolyticus* strains. Using real-time microscopy, we found that the death of BMDMs derived from *Nlrp1*[-/-] mice was comparable to that of BMDMs generated from WT mice. However, infection of *Nlrp3*[-/-] BMDMs resulted in delayed cell death and ~60% reduction in the calculated AUC (*Figure 3a*, *Figure 3—figure supplement 1a*). Furthermore, hallmark processes of inflammasome-mediated cell death, such as IL-1β, Caspase-1, and GSDMD cleavage and release, which were induced in a T6SS3-dependent manner, were abrogated only in *Nlrp3*[-/-] BMDMs (*Figure 3b–d*). TNFα secretion, which was used as a control, was not affected in either of the KO mouse BMDMs (*Figure 3b*), thus confirming that priming (i.e. the NF-κB-dependent pathway) remained unaffected in the *Nlrp3*[-/-] BMDMs.

NLRP3 can also be activated by the necroptotic cell death pathway, which is induced by pseudo-kinase mixed lineage kinase domain-like (MLKL) (*Conos et al., 2017*; *Lawlor et al., 2015*). To confirm that T6SS3 directly activated the NLRP3-induced cell death rather than it being an indirect consequence of a secondary mechanism of necroptotic cell death, we also monitored cell death in *Mlkl*[-/-] BMDMs. As expected, MLKL deficiency did not affect the induction of the T6SS3-mediated cell death (*Figure 3a*, *Figure 3—figure supplement 1a*) or IL-1β, Caspase-1, and GSDMD cleavage and release (*Figure 3b–d*) in BMDMs. Thus, we concluded that T6SS3 activates a pyroptotic cell death program that is mediated directly by the NLRP3 inflammasome.

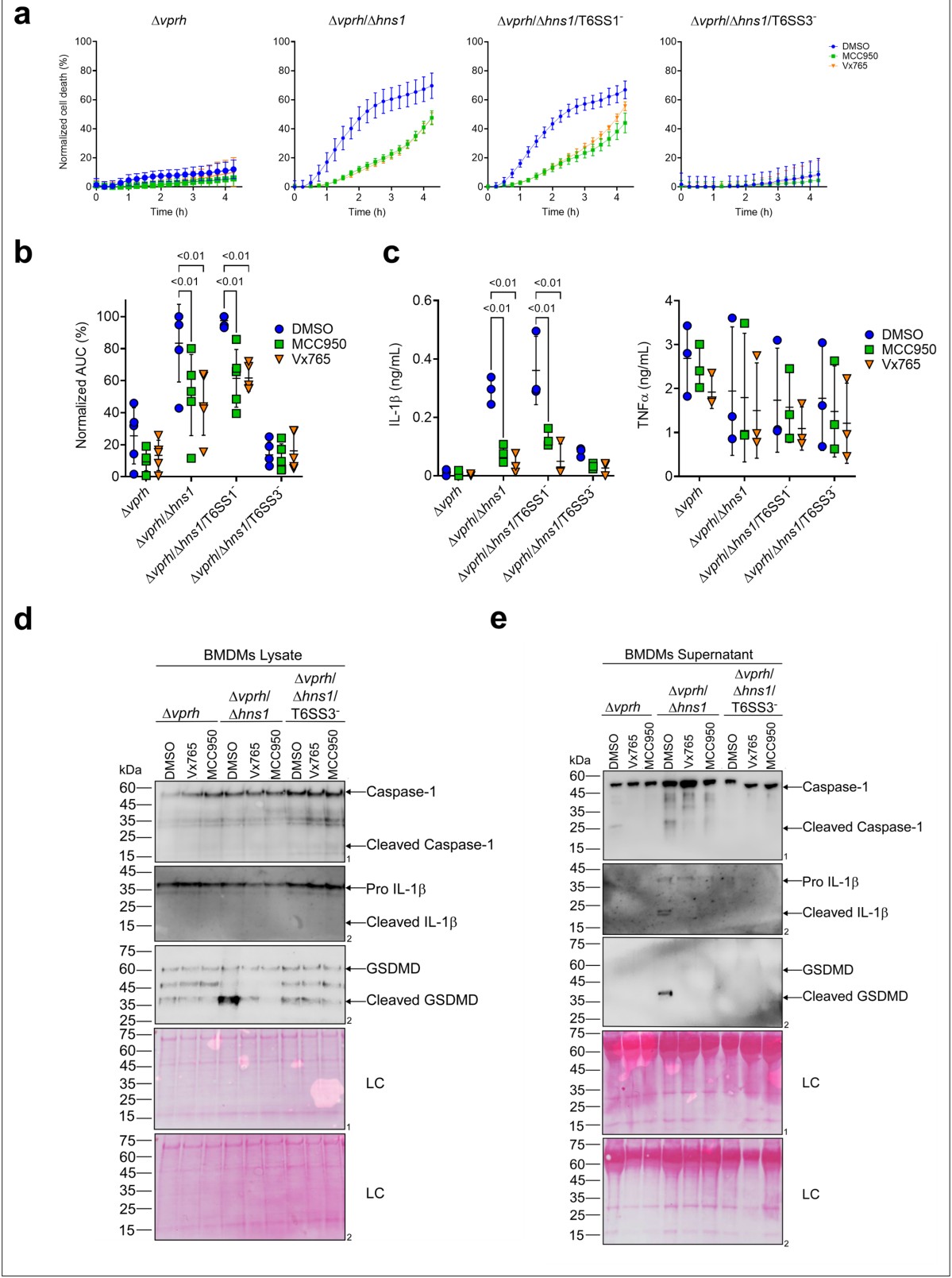

**Figure 2.** T6SS3 induces pyroptotic cell death in BMDMs. Approximately 3.5×10⁴ wild-type bone marrow-derived macrophages (BMDMs) were seeded into 96-well plates in six replicates and were primed using lipopolysaccharides (LPS) (100 ng/mL) for 3 hr prior to infection with *V. proteolyticus* strains at multiplicity of infection (MOI) 5. Where indicated, inflammasome inhibitors Vx765 (25 µM) or MCC950 (2 µM), with the addition of propidium iodide (PI) (1 µg/mL), were added to the cells 30 min prior to bacterial infection. Dimethyl sulfoxide (DMSO) was used as the solvent control. (**a–b**) PI kinetic uptake

*Figure 2 continued on next page*

*Figure 2 continued*

was assessed using real-time microscopy (IncucyteZOOM) (**a**) and then graphed as the area under the curve (AUC) of the percentage of PI-positive cells normalized to the number of cells in the wells (**b**). (**c**) Cell supernatants from experiments described in (**a**) were collected 3 hr postinfection. IL-1β and TNFα secretion were measured using commercial ELISA kits. (**d-e**) Caspase-1, gasdermin D (GSDMD), and IL-1β were detected in BMDM lysate (**d**) and supernatant (**e**) by immunoblotting (the number on the right of each blot denotes the blot number). The data in (**a**) represent n≥3 independent experiments. Statistical comparisons in (**b–c**) between the different treatments were performed using Repeated measures (RM) two-way ANOVA, followed by Turkey's multiple comparison test. The results are shown as the mean ± SD of n≥3 independent experiments; significant differences (p<0.05) are denoted only for comparisons between inflammasome inhibitors of cells infected with the same bacterial strain. The results shown in (**d-e**) represent two independent experiments. Arrows denote the expected band size.

The online version of this article includes the following source data for figure 2:

**Source data 1.** Immunoblotes of Caspase-1, IL-1beta, and GSDMD in BMDM lysates.

**Source data 2.** Immunoblotes of Caspase-1, IL-1beta, and GSDMD in BMDM supernatants.

To confirm that the observed NLRP3-dependent phenotypes are indeed mediated by bacterial T6SS3 activity, we complemented the deletion in *tssL3*, which was used to generate T6SS3⁻ bacterial strains, from an arabinose-inducible plasmid (pTssL3). The complementation of *tssL3* in trans restored the secretion of Hcp3 (*Figure 3—figure supplement 1b*), as well as all of the tested inflammasome-mediated phenotypes, including cell death and IL-1β, Caspase-1, and GSDMD cleavage and release (*Figure 3e and f* & *Figure 3—figure supplement 1c,d*). Importantly, TNFα secretion was not affected upon BMDMs infection with the TssL3-complemented *V. proteolyticus* strain (*Figure 3—figure supplement 1e*). Notably, over-expression of TssL3 had no effect on bacterial growth in MLB media (*Figure 3—figure supplement 1f*), but had a minor positive effect on bacterial growth in DMEM media (*Figure 3—figure supplement 1f*).

## GSDME partially compensates for GSDMD absence in T6SS3-induced pyroptosis and IL-1β secretion

A recent report suggested that in the absence of GSDMD, a cascade comprising Caspase-1, Caspase-8, Caspase-3, and GSDME can lead to cell death in BMDMs (*Tsuchiya et al., 2019*). To investigate whether T6SS3 can induce this cascade, we monitored *Gsdmd⁻/⁻* BMDMs upon infection with *V. proteolyticus* strains. Using real-time microscopy, we found that in contrast with the negligible cell death observed upon infection of *Nlrp3⁻/⁻* BMDMs (*Figure 4a and b*), *V. proteolyticus* strains containing an active T6SS3 (i.e., Δ*vprh*/Δ*hns1* and Δ*vprh*/Δ*hns1*/T6SS1⁻) induced a significant cell death in the absence of GSDMD, albeit delayed, compared to that observed in BMDMs from WT mice. Remarkably, IL-1β secretion from *Gsdmd⁻/⁻* BMDMs infected with the Δ*vprh*/Δ*hns1* or the Δ*vprh*/Δ*hns1*/T6SS1⁻ strain was comparable to that detected in BMDMs generated from WT mice (*Figure 4c*). As before, TNFα secretion was not affected in either of the KO mouse-derived BMDMs (*Figure 4c*). Moreover, while we detected the hallmark pyroptotic phenotypes in infected BMDMs generated from WT mice (i.e. cleaved and released Caspase-1, IL-1β, and GSDMD), in *Gsdmd⁻/⁻* BMDMs we observed the cleavage of Caspase-1, Caspase-3 (but not Caspase-8) and GSDME, in addition to cleavage and release of IL-1β (*Figure 4d and e*). These results suggest that in the absence of GSDMD, T6SS3 can induce a cell death cascade comprising Caspase-1, Caspase-3, and GSDME. Therefore, GSDME can compensate, at least partially, for the absence of GSDMD. Notably, activation of this alternative cascade still requires the NLRP3 inflammasome, since cleaved Caspase-1, Caspase-3 and GSDME were absent in *Nlrp3⁻/⁻*-derived BMDMs infected with the Δ*vprh*/Δ*hns1* or Δ*vprh*/Δ*hns1*/T6SS1⁻ strains (*Figure 4d and e*).

## Activation of T6SS3 by Ats3 is sufficient to induce the NLRP3 inflammasome

The above-mentioned T6SS3-dependent activation of the NLRP3 inflammasome was observed upon infection of BMDMs with bacterial strains harboring a deletion in *hns1*. Since H-NS is a global repressor (*Dorman, 2004*), and thus the deletion of *hns1* may have affected additional cellular systems other than T6SS1 and T6SS3, we set out to identify a positive regulator of T6SS3 that can be used to induce T6SS3, without affecting the activity of T6SS1.

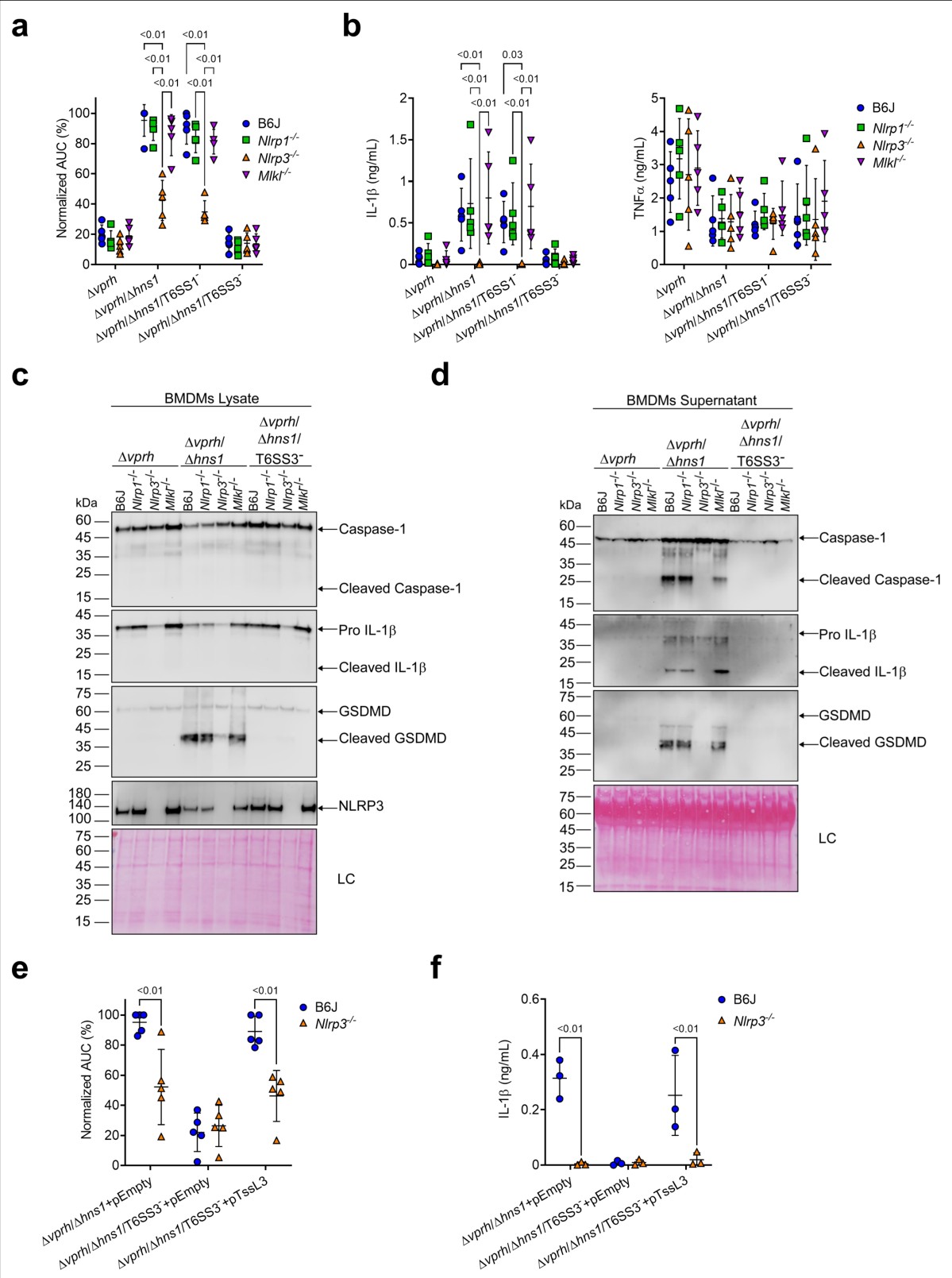

**Figure 3.** T6SS3 activates the NLRP3 inflammasome in BMDMs. Approximately 3.5×10⁴ wild-type (B6J), *Nlrp1⁻/⁻*, *Nlrp3⁻/⁻*, and *Mlkl⁻/⁻* bone marrow-derived macrophages (BMDMs) were seeded into 96-well plates in six replicates and were primed using lipopolysaccharides (LPS) (100 ng/mL) for 3 hr prior to infection with *V. proteolyticus* strains at multiplicity of infection (MOI) 5. In (**e–f**), 0.05% arabinose was added to the media prior bacterial infection. (**a, e**) Propidium iodide (PI) uptake was assessed using real-time microscopy (IncucyteZOOM) and then graphed as the area under the curve

*Figure 3 continued on next page*

*Figure 3 continued*

(AUC) of the percentage of PI-positive cells normalized to the number of cells in the wells. (**b, f**) Cell supernatants from experiments described in (**a**) or (**e**), respectively, were collected 3 hr postinfection. IL-1β and TNFα secretion were measured using commercial ELISA kits. (**c–d**) NLRP3, Caspase-1, gasdermin D (GSDMD), and IL-1β were detected in BMDM lysate (**c**) and supernatant (**d**) by immunoblotting (the number on the right side of each blot denotes the blot number). Arrows denote the expected band size. Statistical comparisons in (**a–b**) and (**e–f**) between the different treatments were performed using RM two-way ANOVA, followed by Turkey's multiple comparison test. The results are shown as the mean ± SD of 5 independent experiments; significant differences (p<0.05) are denoted only for comparisons between mice strains infected with the same bacterial strain. The results in (**c–d**) represent two independent experiments.

The online version of this article includes the following source data and figure supplement(s) for figure 3:

**Source data 1.** Immunoblotes of Caspase-1, IL-1beta, GSDMD, and NLRP3 in BMDM lysates.

**Source data 2.** Immunoblotes of Caspase-1, IL-1beta, and GSDMD in BMDM supernatants.

**Figure supplement 1.** T6SS3 activates the NLRP3 inflammasome in BMDMs.

**Figure supplement 1—source data 1.** Immunoblotes of *V. proteolyticus* Hcp3 expression and secretion.

**Figure supplement 1—source data 2.** Immunoblotes of Caspase-1, IL-1beta, GSDMD, and NLRP3 in BMDM lysates.

**Figure supplement 1—source data 3.** Immunoblotes of Caspase-1, IL-1beta, and GSDMD in BMDM supernatants.

*VPR01S_RS02370* is found at the edge of the T6SS3 gene cluster (*Figure 5a*), and it is transcribed in the opposite direction from its neighboring T6SS3 cluster gene, *VPR01S_RS02375*. *VPR01S_RS02370* encodes a predicted transcription regulator belonging to the GlxA superfamily (according to the NCBI Conserved Domain Database *Lu et al., 2020*). The proximity of *VPR01S_RS02370* to the T6SS3 cluster, and the absence of a recognizable transcription regulator within the cluster led us to hypothesize that this gene, henceforth referred to as *ats3* (activator of type six secretion system 3), regulates T6SS3. To test this hypothesis, we introduced *ats3* on an arabinose-inducible expression plasmid into the Δ*vprh* and Δ*vprh*/T6SS3⁻ strains, and determined its effect on T6SS1 and T6SS3 activity. Importantly, we found that T6SS3 was dramatically induced upon Ats3 over-expression, as evident by elevated expression and secretion of Hcp3, whereas T6SS1 activity (determined by monitoring VgrG1 expression and secretion) remained unaltered (*Figure 5b*). Remarkably, over-expression of Ats3 had a stronger effect on T6SS3 activity than deletion of *hns1* (*Figure 5b*). These results suggest that Ats3 activates T6SS3 and not T6SS1.

Previous work showed that T6SS1 alone was responsible for *V. proteolyticus*'s antibacterial activity during interbacterial competition (*Ray et al., 2017*). To further confirm that Ats3 affects T6SS3 and not T6SS1, we determined whether Ats3 over-expression results in enhanced T6SS1-mediated antibacterial activity. Interbacterial competition assays against sensitive *E. coli* prey (*Ray et al., 2017*) revealed that in contrast to the enhanced antibacterial activity seen upon deletion of *hns1*, which induces both T6SS1 and T6SS3, over-expression of Ats3 had no effect on the antibacterial activity (*Figure 5c*). Furthermore, over-expression of Ats3 did not confer antibacterial activity to a strain in which T6SS1 was inactivated (Δ*vprh*/T6SS1⁻). Taken together, these results indicate that Ats3 is a positive regulator of T6SS3, and that T6SS3 does not play a role in interbacterial competition.

We reasoned that Ats3 can be used to uncouple the effect of T6SS3 on pyroptosis induction from T6SS1 activity. Therefore, to confirm the specific role of T6SS3 in pyroptosis induction, we set out to investigate the effect of Ats3 over-expression on inflammasome activation and cell death. Using real-time microscopy, we found that over-expression of Ats3 from a plasmid (pAts3) induced a rapid, T6SS3-dependent cell death in BMDMs, which was similar to that induced upon *hns1* deletion (*Figure 5d and e* & *Figure 5—figure supplement 1a*). Moreover, infection of *Nlrp3⁻/⁻* BMDMs with Δ*vprh* bacteria carrying pAts3 resulted in reduced cell death compared to that observed upon infection of BMDMs generated from WT mice. In agreement, IL-1β, Caspase-1, and GSDMD cleavage and release were also induced, in a T6SS3-dependent manner, upon Ats3 over-expression (*Figure 5f and g* & *Figure 5—figure supplement 1b*); in contrast, TNFα secretion was not affected by either of *V. proteolyticus* strains in both WT and *Nlrp3⁻/⁻* BMDMs (*Figure 5f*). Notably, Ats3 over-expressing bacteria induced a slow, NLRP3-independent cell death deficient in IL-1β, Caspase-1, and GSDMD cleavage; this result suggests that another cell death mechanism could have been induced upon T6SS3 activation in the absence of NLRP3 (*Figure 5e–g* & *Figure 5—figure supplement 1b*). The apparent cell death in BMDMs is probably not a result of a fitness difference between *V. proteolyticus* strains, since we did not detect any difference in either bacterial growth or motility (*Figure 5—figure*

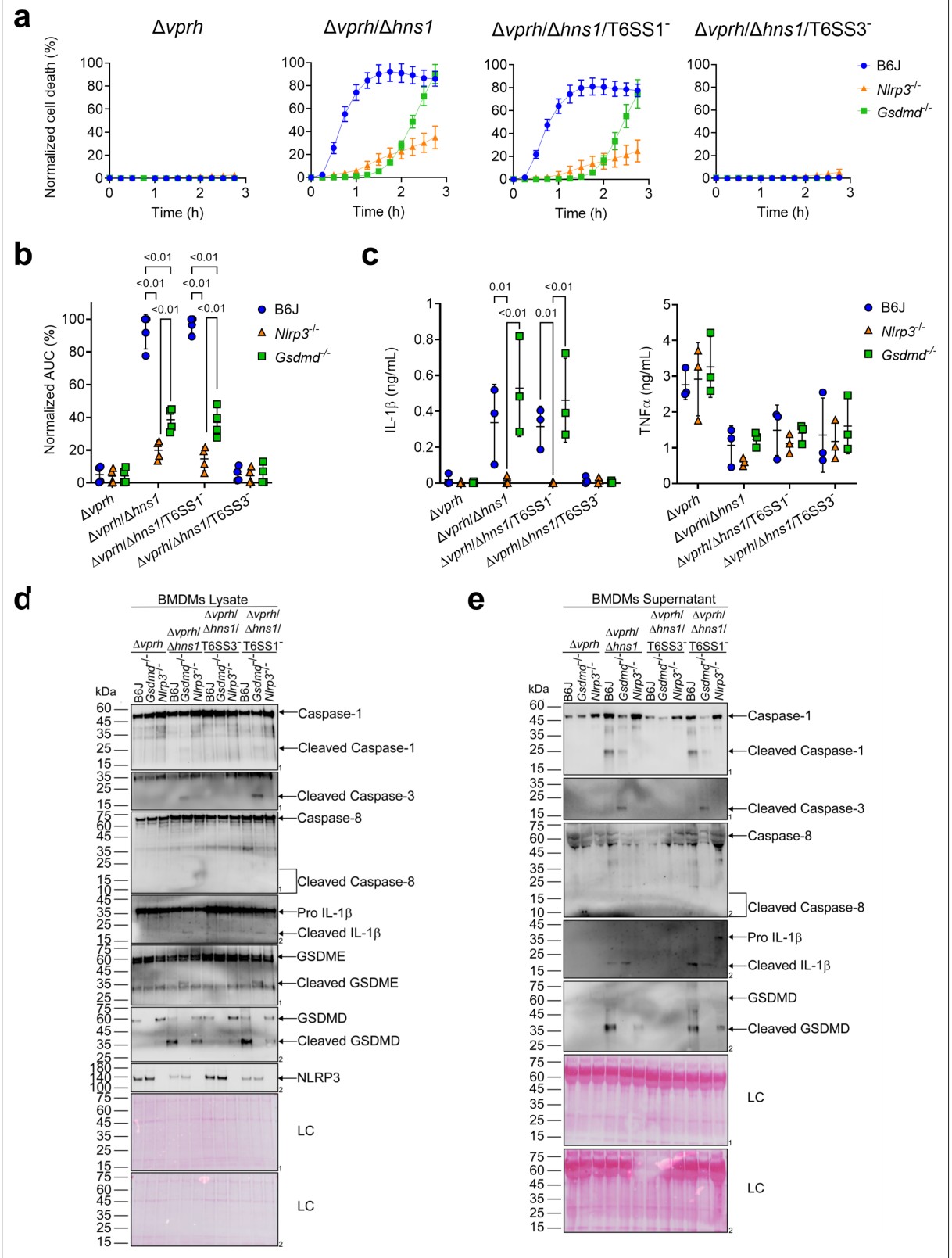

**Figure 4.** GSDME partially compensates for GSDMD absence in T6SS3-induced pyroptosis and IL-1β secretion. Approximately $3.5 \times 10^4$ wild-type (B6J), $Nlrp3^{-/-}$ and $Gsdmd^{-/-}$ bone marrow-derived macrophages (BMDMs) were seeded into 96-well plates in six replicates and were primed using lipopolysaccharides (LPS) (100 ng/mL) for 3 hr prior to infection with *V. proteolyticus* strains at multiplicity of infection (MOI) 5. (**a–b**) PI uptake was assessed using real-time microscopy (IncucyteZOOM) (**a**) and then graphed as the area under the curve (AUC) of the percentage of PI-positive cells

*Figure 4 continued*

normalized to the number of cells in the wells (**b**). (**c**) Cell supernatants described in (**a**) were collected 3 hr post infection. IL-1β and TNFα secretion were measured using commercial ELISA kits. (**d-e**) NLRP3, Caspase-1, Caspase-3, Caspase-8, gasdermin (GSDM)D, GSDME, and IL-1β were detected in BMDM lysate (**d**) and supernatant (**e**) by immunoblotting (the number on the right side of each blot denotes the blot number). Arrows denote the expected band size; denote the expected band sizes for cleaved Caspase-8. The data in (**a**) represent n≥3 independent experiments. Statistical comparisons in (**b–c**) between the different treatments were performed using RM two-way ANOVA, followed by Turkey's multiple comparison test. The results in (**b,c**) are shown as the mean ± SD of n≥3 independent experiments; significant differences (p<0.05) are denoted only for comparisons between mice strains infected with the same bacterial strain. The results shown in (**d-e**) represent two independent experiments.

The online version of this article includes the following source data for figure 4:

**Source data 1.** Immunoblotes of Caspase-1, Caspase-3, Caspase-8, IL-1beta, GSDME, GSDMD, and NLRP3 in BMDM lysates.

**Source data 2.** Immunoblotes of Caspase-1, Caspase-3, Caspase-8, IL-1beta, and GSDMD in BMDM supernatants.

*supplement 1c,d*). Combined, these results confirm that T6SS3 is specifically responsible for the inflammasome-mediated cell death induction in BMDMs.

## T6SS3-induced pyroptosis requires bacterial internalization

Previous reports suggested that for some T6SSs, delivery of anti-eukaryotic effectors requires internalization by a host cell (*Ma et al., 2009*; *Shalom et al., 2007*). This phenomenon was also observed for the *V. proteolyticus* T6SS1-mediated manipulation of the actin cytoskeleton (*Ray et al., 2017*). Therefore, we asked whether the T6SS3-mediated phenotypes observed upon infection of BMDMs are dependent on phagocytosis. Indeed, we found that the addition of cytochalasin D, which inhibits phagocytosis by inhibiting actin polymerization (*Brown and Spudich, 1979*; *Flanagan and Lin, 1980*), abolished the pyroptosis induced by the T6SS3 of a Δ*vprh V. proteolyticus* strain over-expressing Ats3 (*Figure 6a*, *Figure 6—figure supplement 1a*). Interestingly, cytochalasin D also abolished the weaker inflammasome-independent, T6SS3-dependent cell death observed upon infection of *Nlrp3*[−/−] BMDMs. Notably, cytochalasin D did not abolish the cell death induced by T3SS1 of another pathogen, *V. parahaemolyticus*, which is independent of bacterial internalization (*Burdette et al., 2008*; *Park et al., 2004*) and was therefore used as a control (*Figure 6—figure supplement 1b*); it also did not impair bacterial growth (*Figure 6—figure supplement 1c*). Combined, these results suggest that the T6SS3-mediated pyroptotic cell death requires phagocytosis.

Phagocytic cells, such as BMDMs, internalize bacteria to eliminate them. We reasoned that *V. proteolyticus* activate the T6SS3 upon phagocytosis because this system provides a survival advantage. Indeed, when we measured the bacterial load after 3 hr of BMDM infection, we observed a dramatic advantage for *V. proteolyticus* with an active T6SS3 (Δvprh +pAts3; parental strain) over a derivative strain with an inactive T6SS3 (T6SS3[−]) (*Figure 6b*). Notably, bacterial counts were comparable between the two strains in the absence of BMDMs (*Figure 6b*). Remarkably, the presence of BMDMs appeared to provide a growth advantage to bacteria with an active T6SS3 or to bacteria that were not phagocytosed by BMDMs (i.e. in the presence of cytochalasin D), whereas bacteria with an inactive T6SS3 that were phagocytosed (i.e. in the absence of cytochalasin D) grew less than their counterparts in the absence of BMDMs (*Figure 6b*). Taken together, these results demonstrate that T6SS3 provides a survival advantage to *V. proteolyticus* upon its internalization by an immune cell.

## Two T6SS3 effectors are necessary and sufficient to induce pyroptotic cell death

While analyzing the T6SS3 gene cluster, we were intrigued by two genes found adjacent to *ats3*, namely, *VPR01S_RS02380* and *VPR01S_RS02375* (*Figure 5a*). The two genes, hereafter termed *tie1* (T6SS3 inflammasome-inducing effector 1) and *tie2*, respectively, encode proteins with no apparent similarity to known T6SS core or accessory components (*Boyer et al., 2009*). Interestingly, hidden Markov model analyses (using the Jackhmmer server *Potter et al., 2018*) revealed that Tie2 shares partial similarity (in the region between amino acids 326 and 469) with domains found at C-termini of polymorphic toxins, such as proteins containing N-terminal LXG and WXG100 domains that are associated with type VII secretion systems (*Tran et al., 2021*; *Whitney et al., 2017*), as well as DUF4157 domains, which were recently suggested to carry toxins in extracellular contractile injection systems (*Geller et al., 2021*). These observations led us to hypothesize that Tie1 and Tie2 are T6SS3 effectors.

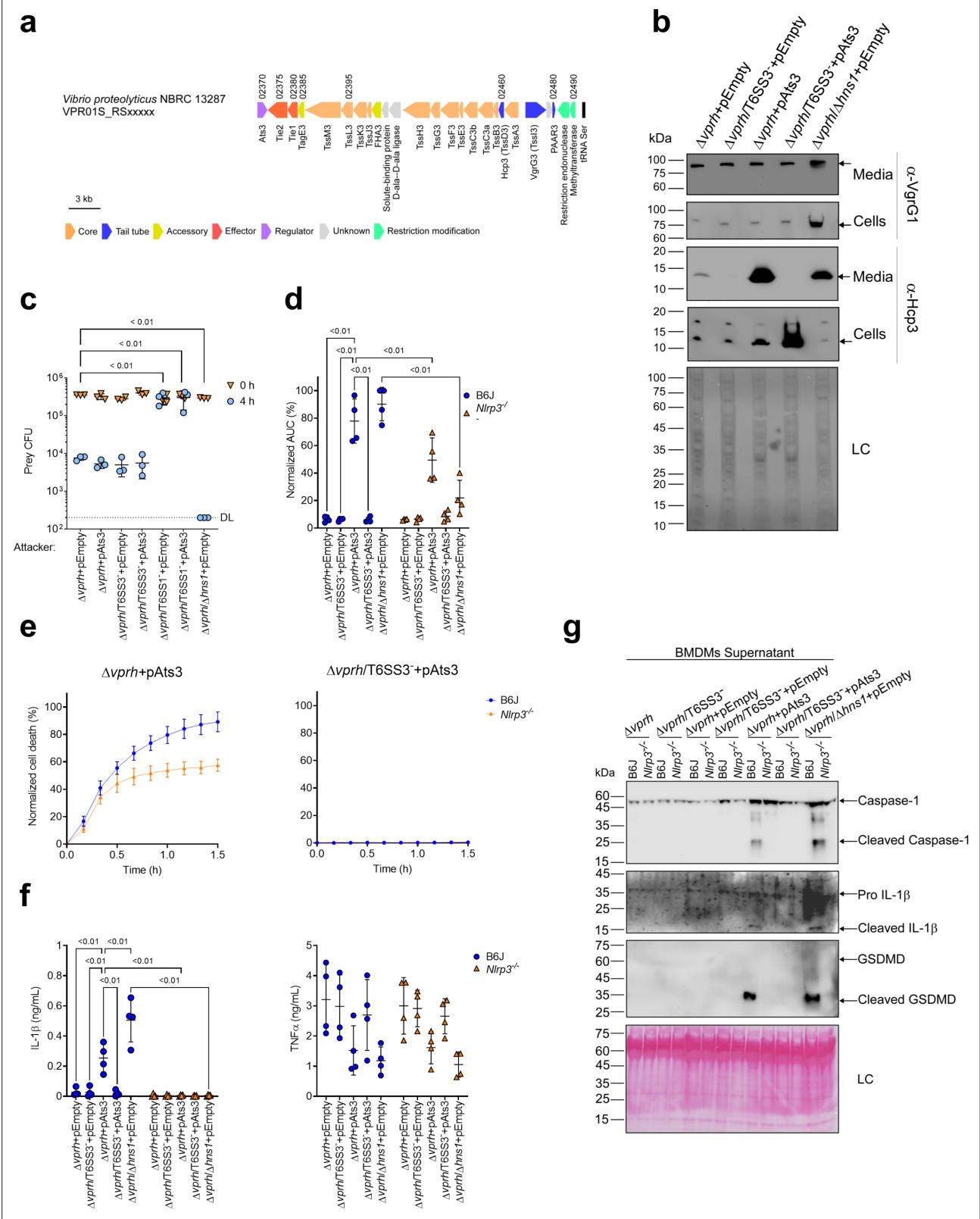

**Figure 5.** Activation of T6SS3 by Ats3 is sufficient to induce the NLRP3 inflammasome. (**a**) The T6SS3 gene cluster. Genes are represented by arrows indicating direction of transcription. Locus tags (*vpr01s_RSxxxxx*) are denoted above; encoded proteins and known domains are denoted below. (**b**) The expression (cells) and secretion (media) of Hcp3 and VgrG1 were detected by immunoblotting using specific antibodies against Hcp3 and VgrG1, respectively. Loading control (LC) is shown for total protein lysate. (**c**) Viability counts of *E. coli* XL-1 blue prey before (0 hr) and after (4 hr) co-incubation

*Figure 5 continued on next page*

Figure 5 continued

with the indicated attackers, on media containing 3% (w/v) NaCl and 0.1% (w/v) arabinose at 30°C. (**d-g**) Approximately $3.5×10^4$ wild-type (B6J) and *Nlrp3*[-/-] bone marrow-derived macrophages (BMDMs) were seeded into 96-well plates in six replicates and were primed using lipopolysaccharides (LPS) (100 ng/mL) for 3 hr prior to infection with *V. proteolyticus* strains at multiplicity of infection (MOI) 5. Arabinose (0.05% w/v) was added to the media prior bacterial infection. (**d-e**) Propidium iodide (PI) uptake was assessed using real-time microscopy (IncucyteZOOM) and then was graphed as the area under the curve (AUC) of the percentage of PI-positive cells normalized to the number of cells in the wells. (**f**) Cell supernatants from experiments described in (**a**) were collected 3 hr postinfection. IL-1β and TNFα secretion were measured using commercial ELISA kits. (**g**) NLRP3, Caspase-1, gasdermin D (GSDMD), and IL-1β were detected in BMDM supernatants by immunoblotting. Arrows denote the expected band size. The data in (**b–c, e**) and (**g**) represent the results of n≥3 and n=2 independent experiment, respectively. The data in (**c**) are shown as the mean ± SD. Statistical comparisons in (**c**) between samples at the 4 hr timepoint were performed using an unpaired, two-tailed Student's *t*-test; significant differences (p<0.05) are denoted above. DL, assay detection limit. Statistical comparisons in (**d**) and (**f**) between the different treatments were performed using RM two-way ANOVA, followed by Turkey's multiple comparison test. The results are shown as the mean ± SD of n≥4 independent experiments; significant differences (p<0.05) are denoted for comparisons between infections of cells from the same mouse strain, and between infections using the same *V. proteolyticus* strain.

The online version of this article includes the following source data and figure supplement(s) for figure 5:

**Source data 1.** Immunoblots of *V. proteolyticus* Hcp3 and VgrG1 expression and secretion.

**Source data 2.** Immunoblotes of Caspase-1, IL-1beta, and GSDMD in BMDM supernatants.

**Figure supplement 1.** Activation of T6SS3 by Ats3 is sufficient to induce the NLRP3 inflammasome.

**Figure supplement 1—source data 1.** Immunoblotes of Caspase-1, IL-1beta, and GSDMD in BMDM lysates.

To test this hypothesis, we generated custom-made antibodies against Tie1 and Tie2, and monitored their expression and secretion upon over-expression of Ats3, which activates T6SS3. As shown in *Figure 7a*, Tie1 and Tie2 were secreted in a T6SS3-dependent manner, as was Hcp3; in contrast, secretion of VgrG1, a T6SS1 component, was unaffected by Ats3 over-expression. These results confirm that both Tie1 and Tie2 are T6SS3 secreted proteins.

The above findings prompted us to posit that if Tie1 and Tie2 are indeed T6SS3 effectors, then they can be responsible for the T6SS3-mediated induction of pyroptosis in BMDMs. To test this hypothesis, we generated *V. proteolyticus* mutant strains with deletions in either *tie1* (deletion of the region corresponding to nucleotides 485–584 in *tie1*), *tie2*, or both; we used them to infect BMDMs, and then we monitored the effect on cell death. Remarkably, deletion of both *tie1* and *tie2* in a Δ*vprh*/Δ*hns1* background (Δ*vprh*/Δ*hns1*/Δ*tie1*/Δ*tie2*) completely abolished the cell death and IL-1β secretion observed upon infection of BMDMs with the Δ*vprh*/Δ*hns1* strain (*Figure 7—figure supplement 1a,b*), without affecting TNFα secretion (*Figure 7—figure supplement 1b*). Deletion of *tie1* alone resulted in an intermediate cell death phenotype, whereas deletion of *tie2* alone had no significant effect. Complementation of Tie1 and Tie2 expression from a plasmid in the Δ*vprh*/Δ*hns1*/Δ*tie1*/Δ*tie2* strain revealed that over-expression of Tie1, either alone (pTie1) or together with Tie2 (pTie1-2), completely restored the cell death phenotype, whereas bacteria over-expressing only Tie2 (pTie2) regained an intermediate ability to induce cell death (*Figure 7b and c*). Similar effects of *tie1* and *tie2* deletions were observed in the tested inflammasome-activation hallmark phenotypes, including IL-1β, Caspase-1, and GSDMD cleavage and release (*Figure 7d–f* & *Figure 7—figure supplement 1b-d*). TNFα secretion was not significantly affected by complementation of Tie1 or Tie2 when BMDMs were infected with the indicated *V. proteolyticus* strains (*Figure 7d*). Importantly, deletion of *tie1*, *tie2*, or both did not affect T6SS3 functionality, as evident by the secretion of Hcp3 (*Figure 7—figure supplement 1e*); it also did not affect bacterial growth or swimming motility (*Figure 7—figure supplement 1f-g*). Taken together, these results indicate that Tie1 and Tie2 are T6SS3 effectors that are necessary to induce inflammasome-mediated cell death in BMDMs. Either protein is also sufficient to induce cell death, although the effect of Tie1 is more pronounced than that of Tie2.

## T6SS3-like systems are found in pathogenic marine bacteria

Since vibrios are known for their ability to share traits and virulence factors via horizontal gene transfer (HGT) (*Le Roux et al., 2015*), we asked whether T6SS3-like clusters are found in other bacteria. Indeed, we identified highly similar T6SS clusters in other marine bacteria, including human pathogens such as *V. parahaemolyticus* and *V. vulnificus* (*Baker-Austin et al., 2018*), and marine animal pathogens such as *Vibrio crassostreae* (*Bruto et al., 2017*) and *Vibrio anguillarum* (*Frans et al., 2011*; *Figure 8*). Interestingly, these T6SS3-like clusters are often adjacent to DNA mobility elements such as

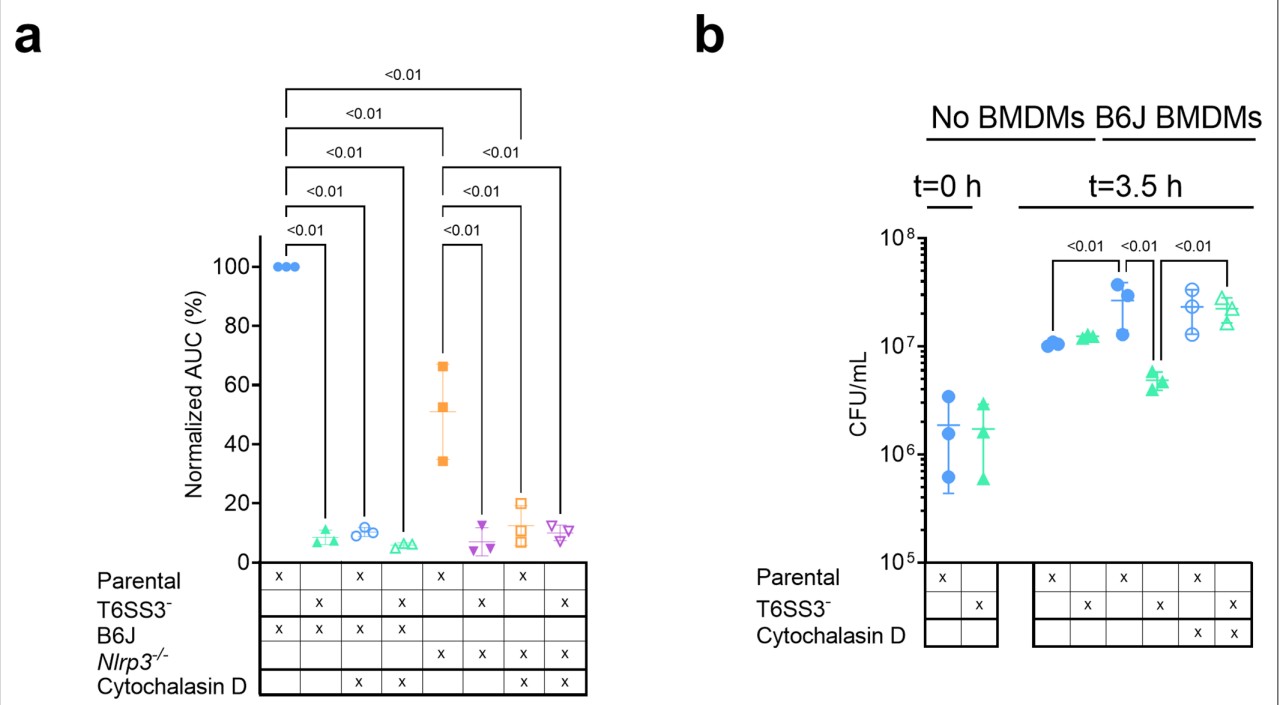

**Figure 6.** *V. proteolyticus* T6SS3-induced pyroptosis requires phagocytosis. Approximately 3.5×10⁴ wild-type (B6J) and *Nlrp3⁻/⁻* bone marrow-derived macrophages (BMDMs) were seeded into 96-well plates in n≥3 replicates and were primed using lipopolysaccharides (LPS) (100 ng/mL) for 3 hr prior to infection with *V. proteolyticus* strains at multiplicity of infection (MOI) 5. Arabinose (0.05% w/v) was added to the media prior to bacterial addition. When indicated, cytochalasin D (5 μM) was added to the cells 45 min prior to bacterial infection. (**a**) Propidium iodide (PI) uptake was assessed using real-time microscopy (IncucyteZOOM) and then graphed as the area under the curve (AUC) of the percentage of PI-positive cells normalized to the number of cells in the wells. (**b**) Bacterial counts of *V. proteolyticus* strains before (t=0 hr) and after (t=3.5 hr) BMDMs infection C. The results in (**a,b**) are shown as the mean ± SD of n=3 independent experiments. Statistical comparisons in (**a**) between the different treatments were performed using RM one-way ANOVA, followed by Sidak multiple comparison test. Statistical comparison in (**b**) between the different treatments were performed using RM one-way ANOVA. Significant differences (p<0.05) are denoted. Parental, *V. proteolyticus* Δvprh +pAts3; T6SS3⁻, *V. proteolyticus* Δvprh/ΔtssL3+pAts3.

The online version of this article includes the following figure supplement(s) for figure 6:

**Figure supplement 1.** *V.proteolyticus* T6SS3 activity requires phagocytosis.

integrases or transposases, suggesting a mechanism for their spread via HGT. Notably, the identified T6SS3-like clusters carry a homolog of at least one of the inflammasome-inducing effectors, Tie1 or Tie2, at the edge of the cluster. Thus, these systems probably maintain their inflammasome-inducing activity, which may contribute to their pathogenic potential.

## Discussion

T6SSs are sophisticated molecular machines that are used by Gram-negative bacteria to inject toxic effector proteins into neighboring cells. Even though most T6SSs studied to date mediate interbacterial competition by injecting antibacterial effectors into neighboring bacteria, few T6SSs were found to target eukaryotes (***Bröms et al., 2010***; ***Clemens et al., 2018***; ***Jiang et al., 2014***; ***Monjarás Feria and Valvano, 2020***; ***Pukatzki et al., 2007***; ***Rosales-Reyes et al., 2012***; ***Sana et al., 2015***; ***Wang et al., 2009***). In this work, we describe T6SS3, a functional T6SS in the marine bacterium *V. proteolyticus*, which induces inflammasome-mediated cell death (known as pyroptosis) upon phagocytosis. We show that this T6SS-mediated pyroptotic cell death is dependent on the delivery at least two novel effectors, and that it involves the activation of the NLRP3 inflammasome, leading to the processing and release of Caspase-1, IL-1β, and GSDMD. Two other T6SS effectors were previously reported to indirectly affect inflammasome activation: EvpP from *Edwardsiella tarda* was shown to inhibit the NLRP3 inflammasome by targeting the MAPK-Jnk pathway (***Chen et al., 2017***), whereas TecA from *Burkholderia cenocepacia* deamidates Rho GTPases, leading to activation of the pyrin-inflammasome

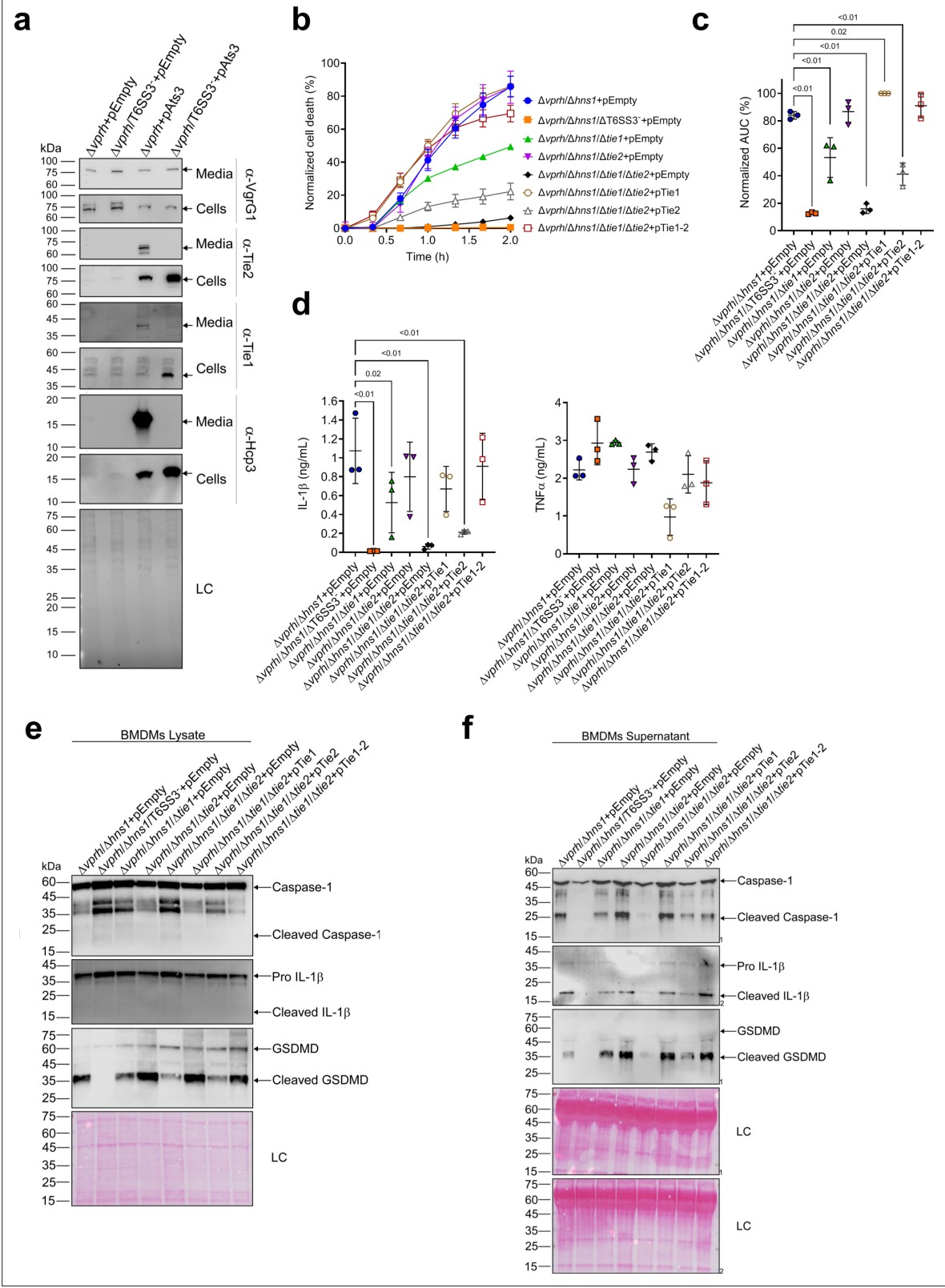

**Figure 7.** Two T6SS3 effectors are necessary and sufficient to induce pyroptosis. (**a**) The expression (cells) and secretion (media) of Tie1, Tie2, Hcp3, and VgrG1 from *V. proteolyticus* strains were detected by immunoblotting using custom-made antibodies. Loading control (LC) is shown for total protein lysate. (**b–f**) Approximately 3.5×10⁴ wild-type bone marrow-derived macrophages (BMDMs) were seeded into 96-well plates in six replicates and were primed using lipopolysaccharides (LPS) (100 ng/mL) for 3 hr prior to infection with *V. proteolyticus* strains at multiplicity of infection (MOI) 5. Arabinose

*Figure 7 continued on next page*

*Figure 7 continued*

(0.05% w/v) and 250 µg/mL kanamycin were added to the media prior to bacterial infection. (**b–c**) Propidium iodide (PI) uptake was assessed using real-time microscopy (IncucyteZOOM) (**b**) and then graphed as the area under the curve (AUC) of the percentage of PI-positive cells normalized to the number of cells in the wells (**c**). (**d**) Cell supernatants from experiments described in (**b**) were collected 3 hr post infection. IL-1β and TNFα secretion were measured using commercial ELISA kits. (**e–f**) NLRP3, Caspase-1, gasdermin D (GSDMD), and IL-1β were detected in BMDM lysates (**e**) and supernatants (**f**) by immunoblotting. The data in (**a–b**) and (**e–f**) represent three and two independent experiments, respectively. Statistical comparisons in (**c–d**) between the different treatments were performed using RM one-way ANOVA, followed by Dunnett's multiple comparison test. The results are shown as the mean ± SD of three independent experiments; significant differences (p<0.05) are denoted only for comparisons of treatments to the Δvprh/Δhns1+pEmpty treatment. In (**a, e, f**), arrows denote the expected band size.

The online version of this article includes the following source data and figure supplement(s) for figure 7:

**Source data 1.** Immunoblots of *V. proteolyticus* Hcp3 expression and secretion.

**Source data 2.** Immunoblots of *V. proteolyticus* Tie1 expression and secretion.

**Source data 3.** Immunoblots of *V. proteolyticus* Tie2 expression and secretion.

**Source data 4.** Immunoblots of *V. proteolyticus* VgrG1 expression and secretion.

**Source data 5.** Immunoblots of *V. proteolyticus* Hcp3, Tie1, Tie2, and VgrG1 expression and secretion.

**Source data 6.** Immunoblotes of Caspase-1, IL-1beta, and GSDMD in BMDM lysates.

**Source data 7.** Immunoblotes of Caspase-1, IL-1beta, and GSDMD in BMDM supernatants.

**Figure supplement 1.** Two T6SS3 effectors are necessary and sufficient to induce pyroptotic cell death.

**Figure supplement 1—source data 1.** Immunoblotes of Caspase-1, IL-1beta, and GSDMD in BMDM supernatans.

**Figure supplement 1—source data 2.** Immunoblots of *V. proteolyticus* Hcp3 and VgrG1 expression and secretion.

---

(*Aubert et al., 2016*). Here, we describe for the first time, to the best of our knowledge, not one but two T6SS effectors that lead to activation of the NLRP3 inflammasome.

Remarkably, we found that in the absence of GSDMD, the canonical gasdermin activated during pyroptosis, an alternative NLRP3-dependent inflammasome cell death cascade, which includes Caspase-1, Caspase-3 (but not Caspase-8) and GSDME, can be induced by this T6SS3. Since IL-1β release is one of the most ancient and conserved immune mechanism (*Dinarello, 2018*), it is impotent to understand how the mammalian immune system evolved to induce several backup mechanisms to ensure its activation. Even in the case of GSDMD inhibition, which may occur if GSDMD is targeted by the pathogen (*Luchetti et al., 2021*), a conserved mechanism including the activation of Caspase-3 and GSDME in a NLRP3 inflammasome-dependent manner will result in the secretion of this proinflammatory cytokine. Notably, a recent study independently reported a similar compensation mechanism in the absence of GSDMD via chemical or *Salmonella*-induced NLRP3 inflammasome activation that also included Caspase-8 activation (*Zhou and Abbott, 2021*). In contrast, we were unable to detect Caspase-8 activation in our system. Therefore, we propose that the cascade revealed in our work involves direct Caspase-3 activation by Caspase-1, as was previously suggested for AIM2 inflammasome in the absence of Caspase-8 (*Sagulenko et al., 2018*).

In the battle between mammalian cells and pathogens, the immune response mechanism that is activated plays a central role in the host's ability to fight-off infections. Inflammasome activation (and pyroptotic cell death), which is an ancient innate immune mechanism, is activated by numerous inputs and cellular stresses (*Lamkanfi and Dixit, 2014*; *Schroder and Tschopp, 2010*). Nevertheless, the immunological consequences of inflammasome activation by pathogens remain enigmatic. Often it is a useful mechanism that enables the host to eliminate infection either by inducing inflammation or by killing infected cells; however, in certain scenarios, inflammasome activation can be exploited by the pathogen, and it provides it with an advantage against the immune system (*Man et al., 2017*). Indeed, we observed a T6SS3-mediated growth advantage to *V. proteolyticus* during BMDMs infection when phagocytosis, which is generally used by the immune cell to eliminate the bacteria, was functional. Future investigations will determine whether T6SS3-induced pyroptosis is beneficial to the bacterium or to the host during in vivo infection.

Although mammals are probably not the natural target for the *V. proteolyticus* T6SS3, nor are they the evolutionary driving force for the activity of the effectors Tie1 and Tie2, immune responses similar to the ones described here (e.g. cell death mechanisms) are found in marine animals that are in direct contact with pathogenic vibrios (e.g. arthropods and fish). Indeed, *Vibrio coralliilyticus* was recently described to activate Caspase-3, leading to cleavage of GSDME and cell death in corals (*Jiang et al.,*

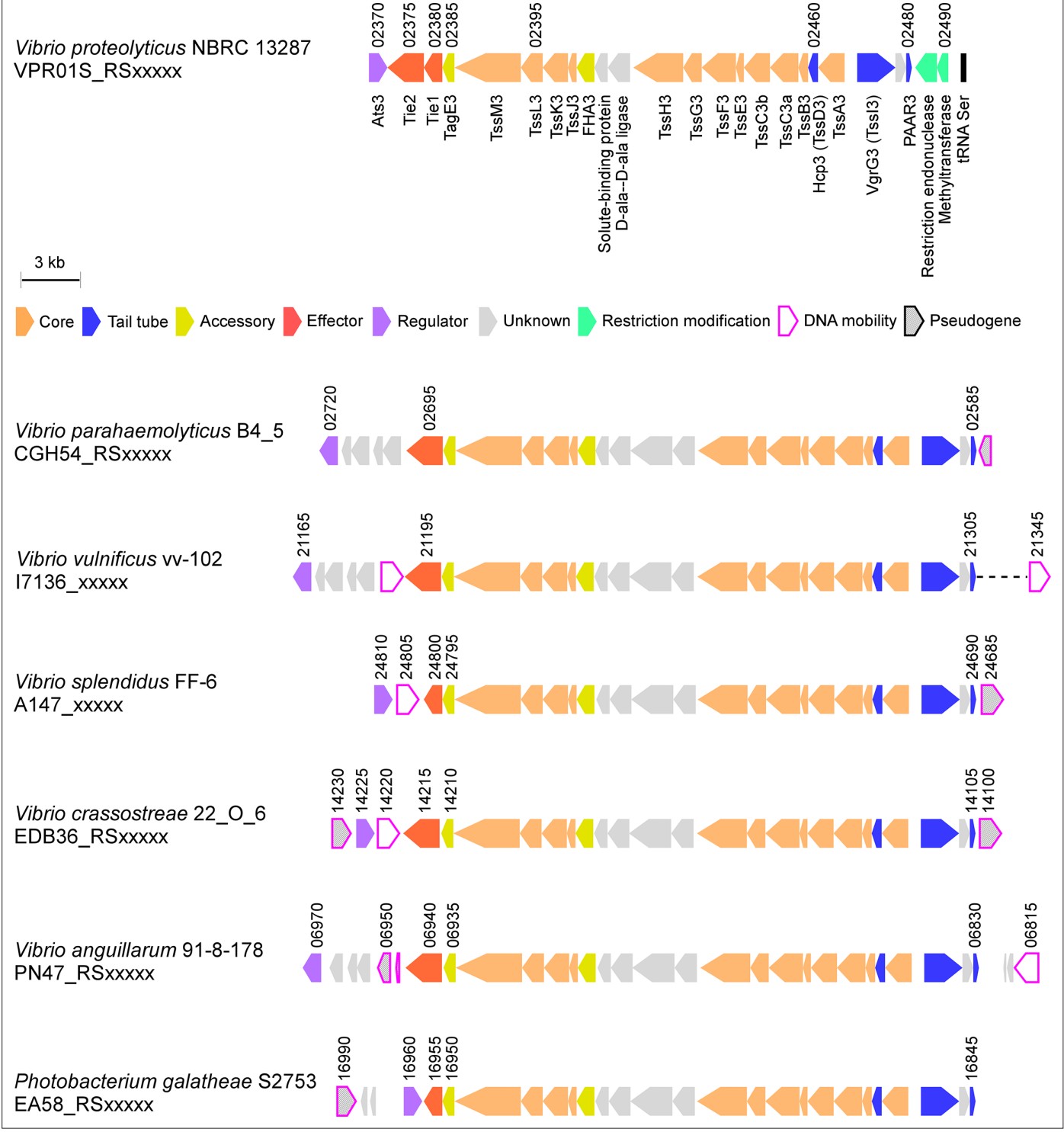

**Figure 8.** T6SS3-like systems are found in pathogenic marine bacteria. T6SS3-like gene clusters. Genes are represented by arrows indicating the direction of transcription. Locus tags are denoted above; encoded proteins and known domains are denoted below for *V. proteolyticus*. A dashed line denotes a gap containing genes that are not shown.

*2020*), while a Caspase-1 homolog was previously described in *Artemia sinica* (*Chu et al., 2014*). Since NLR proteins, such as NLRP3, are less conserved in marine animals, we hypothesize that the target of Tie1 and Tie2 is upstream of NLRP3, and therefore the conservation between mammalian cells and marine animals lies in their ability to sense the effect of Tie1 and Tie2 activity as a danger

signal that activates cell death pathways. Moreover, since we find homologous T6SS clusters that carry Tie1 and Tie2 homologs in other bacteria, including human pathogens, this T6SS and these effectors may contribute to the virulence of some bacteria. Finally, the probable horizontal transfer of this anti-eukaryotic T6SS between marine bacteria may also contribute to the emergence of new pathogenic *Vibrio* strains.

Here, we also shed light on the regulation of T6SS3. We found that similar to T6SS in other bacteria, both T6SS1 and T6SS3 in *V. proteolyticus* are repressed by H-NS. Importantly, we demonstrated that Ats3, which appears to be conserved also in the homologous T6SS clusters found in other bacteria, is an activator of T6SS3 but not of T6SS1. When over-expressed, Ats3 activated T6SS3 even more dramatically than the deletion of *hns1*. Interestingly, the Ats3-induced T6SS3 activated NLRP3 independent cell death in BMDMs, which had different kinetics than the NLRP3-mediated cell death. This result suggests that upon hyper-activation, T6SS3 can induce different cell death mechanisms.

In conclusion, we describe T6SS3, an anti-eukaryotic T6SS in *V. proteolyticus*, and we identify two novel anti-eukaryotic effectors. We also decipher the mechanism of cell death, which is induced by the two effectors in primary macrophages. Nevertheless, future studies will address the question regarding the activity and target of the two effectors, and they will determine how these effectors activate the inflammasome. Moreover, it remains to be determined whether inflammasome-induced cell death is beneficial to the host or to the bacterium during infection.

## Materials and methods

### Reagents

Unless otherwise stated, all cell culture reagents were purchased from Biological Industries, Beit-Haemek, Israel. Lipopolysaccharides (LPS) of *Escherichia coli* O111:B4 were purchased from Sigma-Aldrich (#L3024). Propidium iodide (PI) was purchased from Sigma-Aldrich (#P4170). Vx765 and MCC950 and ELISA kits were purchased from Invitrogen. HRP-conjugated secondary antibodies were purchased from Jackson ImmunoResearch Labs (West Grove, PA, USA). Cytochalsin D (1233) was purchased from Tocris.

### Mice

C57BL/6 (wild-type [WT]), *Nlrp3*$^{-/-}$ (NLRP3tm1Bhk/J), *Mlkl*$^{-/-}$, *Nlrp1*$^{-/-}$ and *Gsdmd*$^{-/-}$ mice were bred under specific pathogens free conditions in the animal facility at Tel Aviv University. All mice are generated on a C57BL/6 background. Experiments were performed on 6–8 weeks old either female or male mice according to the guidelines of the Institute's Animal Ethics Committees.

### Cell culture

Bone marrow (BM) cells from 6 to 8 weeks old either female or male mice were isolated by flushing femurs and tibias with 5 mL PBS supplemented with 2% (v/v) heat-inactivated fetal bovine serum (FBS) Gibco (Thermo Fisher Scientific, Waltham, MA, USA). The BM cells were centrifuged for 5 min at 400 × *g* and then resuspended in DMEM (Sartorius, 01-052-1A) supplemented with 10% (v/v) FBS and 15% L929 conditional medium (L-con). BMDMs were obtained by 7 days differentiation as previously described (*Trouplin et al., 2013*).

### Bacterial strains and media

For a complete list of strains used in this study, see *Supplementary file 1*. *V. proteolyticus* and its derivatives, as well as *V. parahaemolyticus*, were grown in MLB (Lysogeny broth supplemented with NaCl to a final concentration of 3% [w/v]) or on MLB agar plates (supplemented with 1.5% [w/v] agar) at 30°C. Media were supplemented with kanamycin (250 µg/mL) or chloramphenicol (10 µg/mL) when appropriate to maintain plasmids. *E. coli* were grown in 2xYT broth (1.6% [w/v] tryptone, 1% [w/v] yeast extract, and 0.5% [w/v] NaCl) or Lysogeny broth (LB) at 37°C. Media were supplemented with kanamycin (30 µg/mL) or chloramphenicol (10 µg/mL) when appropriate to maintain plasmids. To induce the expression of genes from pBAD plasmids, 0.1% (w/v) L-arabinose was included in the media.

### Plasmid construction

For a complete list of plasmids used in this study, see *Supplementary file 2*. Primers used for amplification are listed in *Supplementary file 3*. For arabinose-inducible expression, the coding sequences

(CDS) of *tssL3*, *ats3*, *tie1*, and *tie2* were amplified from *V. proteolyticus* genomic DNA. Amplicons were inserted into the multiple cloning site (MCS) of pBAD/Myc-His$^{Kan}$ using the Gibson-assembly method (*Gibson et al., 2009*). The constructed plasmids were transformed into *E. coli* DH5 α( $\lambda$ -pir) competent cells using electroporation. Plasmids were conjugated into *V. proteolyticus* using tri-parental mating. Trans-conjugants were selected on MLB agar plates supplemented with appropriate antibiotics to maintain the plasmids.

## Construction of bacterial deletion strains

For in-frame deletions of *V. proteolyticus* genes, 1 kb sequences upstream and downstream of each gene or region to be deleted were cloned into pDM4, a CmROriR6K suicide plasmid (*O'Toole et al., 1996*) using restriction digestion and ligation. These pDM4 constructs were transformed into *E. coli* DH5α ( $\lambda$ -pir) by electroporation, and then transferred into *V. proteolyticus* via conjugation. Trans-conjugants were selected on MLB agar plates containing chloramphenicol (10 µg/mL). The resulting trans-conjugants were grown on MLB agar plates containing sucrose (15% [w/v]) for counter-selection and loss of the SacB-containing pDM4. Deletion was confirmed by PCR.

## Bacterial growth assays

Overnight-grown cultures of *V. proteolyticus* were normalized to an $OD_{600} = 0.01$ in MLB media and transferred to 96-well plates (200 µL per well). For each experiment, n = 3. Cultures were grown at 30 or 37 °C in a BioTek EPOCH2 microplate reader with continuous shaking at 205 cpm. $OD_{600}$ readings were acquired every 10 min. Experiments were performed at least three times with similar results.

## Bacterial swimming assays

Swimming media plates were prepared with Lysogeny broth containing 20 g/L NaCl and 3 g/L Agar. When necessary to induce the expression of genes from a plasmid, 0.1% (w/v) L-arabinose was included in the media. *V. proteolyticus* strains that were grown overnight on an MLB plate were picked and then stabbed into the swimming plates using a toothpick (n = 3). Plates were incubated at 30°C for 8–16 hr. Swimming was assessed by measuring the diameter of the spreading bacterial colony. The experiments were performed three times with similar results.

## Infection experiments

BMDMs were washed three times using PBS and then seeded at a final concentration of 3.5×10⁴ cells/ mL in triplicate in 1% FBS and penicillin–streptomycin-free DMEM. BMDMs were pre-incubated with LPS (100 ng/mL, 3 hr), and then infected with *V. proteolyticus* at MOI 5. When used, inflammasome inhibitors Vx765 (25 µM) and MCC950 (2 µM) were added 30 min prior to infection. For phagocytosis inhibition assay, cytochalasin D (final concentration 5 µM) was added 30 min prior to infection. More specifically, overnight cultures of *V. proteolyticus* strains were washed and normalized to $OD_{600} = 0.016$ (5 MOI) in DMEM without antibiotics. Bacteria were added to wells containing the BMDMs, and plates were centrifuged for 5 min at 400 × *g*. Plates were inserted into the IncucyteZOOM (Essen BioScience) for incubation at 37°C and for monitoring cell death, as detailed below.

## Live cell imaging

Plates containing BMDMs were placed in IncucyteZOOM and images were recorded every 10–30 min. The data were analyzed using the IncucyteZoom2016B analysis software and then exported to the GraphPad Prism software. Normalization was performed according to the maximal PI-positive object count to calculate the percentage of dead cells (*Isherwood et al., 2011*).

## Immune response immunoblot analyses

Cells were collected and pelleted by centrifugation for 5 min at 400 × *g* (4°C). Next, the cells were lysed by adding denaturing (2×) Tris-Glycine SDS Sample Buffer supplemented with 5% (v/v) β-mercaptoethanol. Lysates were loaded onto any-kD gradient ExpressPlus Page precast gels (GenScript). Proteins were transferred onto a nitrocellulose membrane (Bio-Rad), and Ponceau S staining was performed routinely to evaluate the loading accuracy. Membranes were blocked with 5% (w/v) skim milk in Tris-Buffered Saline (TBS) for 1–2 hr, and then probed overnight with primary antibodies (all antibodies were diluted 1:1000, unless noted otherwise): mouse-NLRP3 (AdipoGen; cryo-2), pro and

mature mouse-IL-1β (R&D Systems; AF-401-NA), pro and cleaved mouse Caspase-1 (Santa Cruz; sc-514) (Adipogen; AG-20B-0042-C100), pro and cleaved mouse-GSDMD (Abcam; ab209845), pro and cleaved mouse-GSDME (Abcam, ab215191), cleaved Caspase-3 (Cell Signaling, 9661 S), and Caspase-8 (R&D; AF1650). Relevant horseradish peroxidase-conjugated secondary antibodies were applied for at least 1 hr. Membranes were washed four times in TBS containing 0.1% (v/v) Tween 20 (TBST) between antibody incubations. Antibodies were diluted in TBST containing 5% (w/v) skim milk. Immunoblots were visualized using an ECL kit (Bio-Rad) in an ODYSSEY Fc (Li-COR) equipped with Image Lab software. All images were cropped for presentation; full-size images will be presented upon request.

## Protein secretion assays

*V. proteolyticus* isolates were grown overnight in MLB broth supplemented with antibiotics to maintain plasmids, if needed. Cultures were normalized to $OD_{600} = 0.18$ in 5 mL MLB with appropriate antibiotics and 0.05% (w/v) arabinose, when required. Cultures were grown for 5 hr at 30°C. After 5 hr, for expression fractions (cells), 0.5 $OD_{600}$ units were collected, and cell pellets were resuspended in (2×) Tris-Glycine SDS sample buffer (Novex, Life Sciences). For secretion fractions (media), culture volumes equivalent to 10 $OD_{600}$ units were filtered (0.22 μm), and proteins were precipitated using deoxycholate and trichloroacetic acid (*Bensadoun and Weinstein, 1976*). Cold acetone was used to wash the protein precipitates twice. Then, protein precipitates were resuspended in 20 μL of 10 mM Tris-HCl pH = 8, followed by the addition of 20 μL of (2×) Tris-Glycine SDS Sample Buffer supplemented with 5% (v/v) β-mercaptoethanol. Next, 0.5 μL of 1 N NaOH was added to maintain a basic pH. Expression and secretion samples were boiled and then resolved on any-kD gradient Mini-PROTEAN or CriterionTGX Stain-Free precast gels (Bio-Rad). Expression and secretion were evaluated using western blot with specific, custom-made antibodies against VgrG1 (described previously *Li et al., 2017*), and Hcp3, Tie1, or Tie2 (polyclonal antibodies raised in rabbits against peptides: CQKHNYELEGGEIKD, CVNIGK-KYTDFTEDEL, and STPLGKAVDIPVEKC, respectively). Tie2, Hcp3 and VgrG1 antibodies were used at 1:1000 dilution, and Tie1 antibodies were used at 1:5000 dilution. Protein signals were visualized in a Fusion FX6 imaging system (Vilber Lourmat) using enhanced chemiluminescence (ECL) reagents. Equal loading was assessed using trihalo compounds' fluorescence of the immunoblot membrane.

## Bacterial competition assays

Attacker and prey strains were grown overnight in appropriate broth (MLB for *V. proteolyticus* and 2xYT for *E. coli*) with the addition of antibiotics when maintenance of plasmids was required. Competition assays were performed as previously described (*Salomon et al., 2013*). Briefly, bacterial cultures were normalized to $OD_{600} = 0.5$ and were mixed at a 4:1 ratio (attacker:prey). Triplicates of mixtures were spotted (25 μL) on MLB agar plates containing 0.1% (w/v) arabinose, and incubated for 4 hr at 30°C. Prey colony forming units (CFU) were calculated after the cultures from t=0 hr and t=4 hr were collected and grown on selective media plates. The assay was performed three times with similar results, and the results from representative experiments are shown.

## Bacterial count quantification

BMDM infection experiments were performed as described above. Bacterial counts were assessed at the time of infection (t=0 hr) and 3.5 hr postinfection (t=3.5 hr). To recover bacteria, triton X-100 was added directly into the experiment wells to a final concentration of 1%, and the plate was incubated for 15 min at 37°C. The media were collected from the wells, and tenfold serial dilutions were spotted onto selective media plates. CFU counts were determined after overnight incubation of the plates at 30°C. The assay was preformed three times with similar results.

## Identification of T6SS3-homologous clusters

T6SS3-like clusters were identified by searching for homologs of the *V. proteolyticus* TssM3 protein sequence using BLAST (*Altschul et al., 1990*). The genomic neighborhoods of randomly selected homologs were then manually examined, and representative clusters with similar genetic composition were chosen for presentation.

## Statistical analysis

Date were analyzed using GraphPad prism 9. Data are presented as the mean ± SD. Comparisons was performed using RM one-way ANOVA, followed by Sidak's multiple comparison test or RM two-way

ANOVA, followed by Tukey's multiple comparison test, unless otherwise is indicated. Statistical significance was considered at $p < 0.05$.

## Resource availability

### Lead contact
Further information and requests for resources and reagents should be directed to and will be fulfilled by the lead contact, Motti Gerlic (mgerlic@tauex.tau.ac.il).

### Materials availability
Materials are available from the authors upon reasonable request.

## Acknowledgements

This work was performed in partial fulfillment of the requirements for a Ph.D. degree for (HC), the Sackler Faculty of Medicine, Tel Aviv University, Israel. CMF was supported by a scholarship from the Clore Israel Foundation, a scholarship for outstanding doctoral students from the Orthodox community (The Council for Higher Education), and by a fellowship from the Manna Center Program in Food Safety and Security at Tel Aviv University. The research of DS and MG were supported by the Israel Science Foundation (ISF) (grants 920/17 and 2174/22), and the Recanati Foundation (TAU).

## Additional information

### Funding

| Funder | Grant reference number | Author |
|---|---|---|
| Israel Science Foundation | 2174/22 | Motti Gerlic |
| Israel Science Foundation | 920/17 | Dor Salomon |
| Tel Aviv University Recanati | | Dor Salomon Motti Gerlic |
| Clore Israel Foundation | | Chaya Mushka Fridman |
| Tel Aviv University | Manna Center Program | Chaya Mushka Fridman |

The funders had no role in study design, data collection and interpretation, or the decision to submit the work for publication.

### Author contributions
Hadar Cohen, Conceptualization, Formal analysis, Investigation, Methodology, Writing - original draft, Writing - review and editing; Noam Baram, Validation, Investigation, Methodology; Chaya Mushka Fridman, Validation, Investigation, Visualization, Methodology; Liat Edry-Botzer, Validation, Investigation; Dor Salomon, Motti Gerlic, Conceptualization, Resources, Supervision, Funding acquisition, Investigation, Methodology, Writing - original draft, Writing - review and editing

### Author ORCIDs
Hadar Cohen (iD) http://orcid.org/0000-0002-6715-5539
Dor Salomon (iD) http://orcid.org/0000-0002-2009-9453
Motti Gerlic (iD) http://orcid.org/0000-0001-9518-1833

### Ethics
This study was performed in strict accordance with the recommendations in the Guide for the Care and Use of Laboratory Animals of the National Institutes of Health. All of the animals were handled according to approved institutional animal care and use committee (IACUC) protocols of Tel Aviv University. The protocol was approved by the Committee on the Ethics of Animal Experiments of Tel Aviv University (Permit Number: 01-20-072). Experiments were performed according to the guidelines of the Institute's Animal Ethics Committees.

Decision letter and Author response
Decision letter https://doi.org/10.7554/eLife.82766.sa1
Author response https://doi.org/10.7554/eLife.82766.sa2

## Additional files

### Supplementary files
- MDAR checklist
- Supplementary file 1. A list of bacterial strains used in this study.
- Supplementary file 2. A list of plasmids used in this study.
- Supplementary file 3. A list of primers used in this study.

### Data availability
All data generated or analysed during this study are included in the manuscript and supporting files.

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
