## [Editor Report]

This study reports the function of novel effectors of the type VI secretion system (T6SS) of *Vibrio proteolyticus*, a *Vibrio* isolated from corals. The significance of the findings also extends to the identification of the mechanism by which these effectors induce pyroptotic cell death in mammalian host immune cells. Overall, the reported findings further contribute to the understanding of virulence mechanisms of bacterial pathogens.

---

## [Decision Letter]

[Editors' note: this paper was reviewed by Review Commons.]

---

## [Author Response]

Reviewer #1:Minor edits1. Line 91. Is a bit misleading to say "many other vibrios" possess T3SS. This conveys that this is perhaps the majority, but T3SS in vibrios is at best 50/50. I think best just to delete this sentence.

We deleted this comment, as suggested.

2. Line 102. Revised to "Thus, in this study, we set out to…" Since the entire paragraph starts with "recent study" I missed that this was summary of new data rather than preview of new results.

The sentence was revised as suggested.

3. Line 503. Correct "xxx-584" or more detail on what this means.

*We Thank the reviewer for pointing out this typo.* This refers to the deletion made in tie1, in the region corresponding to nucleotides 485-584 of this gene. The text was corrected accordingly.

4. Line 603. Salmonella should be italicized.

Corrected.

5. Figures. The labelling of the figures is pretty complicated with the long genetic designations. Is it reasonable to for example name the ∆vprh/∆hns1 strain with an abbreviation (such as ∆VH)? Or instead create a strain name, common used approaches would be HC## (for Hadar Cohen) or TAU# for Tel Aviv University. If you go this route, be sure to update the strain list. The current method can be followed, the figures are just complicated.

We thank the reviewer for raising this concern. We acknowledge the difficulty in following the many different strains and mutations. Nevertheless, after considering the proposed modifications to the strain names, we believe that they will not add much clarity, and may even cause some confusion. Therefore, we respectfully decided to keep the current nomenclature in place.

Reviewer #2:Minor edits1. The authors used a hyperactive T6SS (HNS mutant) to investigate its toxicity. Would the authors be able to use a wild type strain to reproduce the function of T6SS?

We have yet to reveal the external cues that lead to full activation of T6SS3 in vitro. Therefore, in the current study we used genetic tools, such as *hns* deletion or Ats3 over-expression, to monitor the effect of this system on immune cells. We will dissect the activating conditions in future studies, but we believe that the use of genetic tools should not affect the validity of the results in the current study, nor their timely publication.

2. The authors showed that Tie1 and Tie2 are secreted by T6SS3. It is important to show if they are actually delivered into the host cells during infection. Otherwise it is hard to conclude that they are truly effectors. The primary concern is the lack of in vivo studies to show that Tie1 and Tie2 are actually effectors that play a role in activation of NLRP3 inflammasome.

We present 3 pieces of evidence that, when taken together, support the conclusion that Tie1 and Tie2 are T6SS3 effectors: (1) the proteins are secreted in a T6SS3-dependent manner; (2) their deletion does not hamper overall T6SS3 activity; and (3) their deletion causes the same loss of NLRP3-mediated inflammasome activation and pyroptosis as does inactivation of T6SS3 by deletion of its structural component, *tssL3*. Although we agree with the reviewer that directly showing delivery of Tie1 and Tie2 into host cells will further strengthen our conclusion, such experiments are quite challenging and difficult to interpret, especially with T6SS effectors that can use diverse mechanisms for secretion through the system. This point was also noted by reviewer #3: “…I believe they were suggesting to demonstrate secretion in host cells. Although this would be nice, it is non-standard and technically not feasible. These types of experiments require genetically fusing the effector with either an enzymatic moiety (e.g. Β lactamase) or fragment of split GFP. Although such approaches have been previously performed, they often result in either blocked or aberrant secretion due to the presence of the added fragment."

Regarding the reviewer’s comment on the lack of in vivo studies: we agree that these are extremely important, yet they are beyond the scope of the current work, as concurred by reviewers #1 and #3:

Reviewer#1 with regard to Reviewer#2: "I don't think mouse (or aquatic animal) studies are essential for this study. The work contributes nicely to our understanding molecular mechanisms of this T6SS system. As noted in my review, there are many additional lines of study that can be pursued from this work, including animal studies, but this should not preclude publication of this work that is itself an intact unit."

Reviewer#3 regarding reviewer #1's comment on Reviewer#2: "I don't believe that reviewer #2 was suggesting to perform mouse or aquatic animal studies by suggesting in vivo demonstration of secretion…”

Reviewer #3:Major comments:1. If the authors believe that GSDME partially compensates in the absence of GSDMD, have they infected a GSDME/GSDMD double knockouts to see if there is an additive effect?

Indeed, this is a very interesting and specific question for the cell death field. We do not currently possess such a GSDME/GSDMD double knockout mouse, and generating one will be a long endeavor. Since its absence does not diminish the importance or the conclusions of the current work, we think that it should not warrant a delay in publication. We do plan to address this question in future studies.

2. It is clear that Ats3 regulates T6SS3, but not the T6SS1; however, there no evidence suggesting that Atg3 does not regulate other gene clusters. For example, have the authors performed RNA seq to compare the transcriptomes of WT and an Ats3 mutant? If not, the authors should refrain using the words "specific activation".

We thank the reviewer for this important note. Indeed, we lack additional data indicating that Ats3’s effect is indeed restricted only to T6SS3. Therefore, we modified the text accordingly and removed mentions of specific T6SS3 activation.

3. In figure 6B, it's unclear why the bacteria infecting cytochalasin D-treated cells grow more than the T6SS3 mutants in the absence of cytochalasin D.

The difference probably stems from the fact that phagocytosis, the major mechanisms by which BMDMs kill bacteria, is hampered in the presence of cytochalasin D, thus allowing bacteria to grow more than when the BMDMs phagocytose them. The results show that in the absence of cytochalasin D, an active T6SS3 counteracts the killing effect by BMDMs with functional phagocytosis.

Minor comments:4. Figure 1A and other secretion assays: The Western blots include loading control (LC) blots. These are non-standard, non-informative, and not required with the inclusion of the western blots on the "cells" fraction. I would suggest removing these as they may confuse the reader.

We respectfully disagree. Loading controls are standard in bacterial secretion assays, and they are important since they confirm comparable loading and allow proper analysis of the results, especially since we aim to determine whether certain mutations affect the expression of T6SS components. Notably, some groups choose to blot for a cytoplasmic protein (e.g., RpoB in Allsop et al., PNAS, 2017; Liang et al., PLoS Pathogens, 2021) instead of showing overall loaded proteins, as shown in our figures.

5. Line 503: "xxx" should reflect the actual nucleotide nubmers

*We thank the reviewer for pointing out this typo.* This refers to the deletion made in tie1, in the region corresponding to nucleotides 485-584 of this gene. The text was corrected accordingly.

6. Since *V. proteolyticus* is an aquatic pathogen, have the authors tried to infect corals, fish, and crustaceans (or derived cells) with WT and effector mutants?

This is an interesting point, and indeed we are setting up such systems and we plan to perform such experiments in the future as part of follow up projects. However, these in vivo studies are beyond the scope of the current manuscript, as also noted by the reviewer in the cross-consultation comments: “…my previous comment on infecting aquatic animals or cells derived from them is non-standard and not necessary…”

7. Are the targeted host proteins in this study (performed with murine BMDM) conserved in the natural hosts for *V. proteolyticus*?

We hypothesize that the conservation is not in the pathway components that are activated upon infection, but rather in the ability of the host cell to sense danger (i.e., to sense the effect of T6SS3 effectors on the host cell or one of its components), which is the role of the NLRP3 inflammasome in mammalian cells. It is well documented that major differences in immune mechanisms exist between mammals and the potential natural marine hosts of *V. proteolyticus* (e.g., corals, arthropods, and fish); therefore, the conservation at the protein level is low. Nevertheless, basic signaling pathways, such as programed cell death, are conserved between the different phyla. For example, a caspase-1 homolog which was found in arthropods (Chu, B. et al. PLoS One (2014). doi:10.1371/journal.pone.0085343) probably induces an apoptotic-like cell death mechanism, similar to apoptosis in *C. elegans*. We now provide further discussion on this point in the text (lines 648-659).